# On the Learnability of Multilabel Ranking

**Vinod Raman**[*]
Department of Statistics
University of Michigan
Ann Arbor, MI 48104
vkraman@umich.edu

**Unique Subedi**[*]
Department of Statistics
University of Michigan
Ann Arbor, MI 48104
subedi@umich.edu

**Ambuj Tewari**
Department of Statistics
University of Michigan
Ann Arbor, MI 48104
tewaria@umich.edu

## Abstract

Multilabel ranking is a central task in machine learning. However, the most fundamental question of learnability in a multilabel ranking setting with relevance-score feedback remains unanswered. In this work, we characterize the learnability of multilabel ranking problems in both batch and online settings for a large family of ranking losses. Along the way, we give two equivalence classes of ranking losses based on learnability that capture most losses used in practice.

## 1   Introduction

*Multilabel ranking* is a supervised learning problem where a learner is presented with an instance $x \in \mathcal{X}$ and is required to output a ranking of $K$ different labels in decreasing order of relevance to $x$. This is in contrast with *multilabel classification* where given an instance $x \in \mathcal{X}$, the learner is tasked with predicting a subset of the $K$ labels without any explicit ordering. Multilabel ranking is a canonical learning problem with a wide range of applications to text categorization, genetics, medical imaging, social networks, and visual object recognition [Joachims, 2005, Schapire and Singer, 2000, McCallum, 1999, Clare and King, 2001, Baltruschat et al., 2019, Wang and Sukthankar, 2013, Bucak et al., 2009, Yang et al., 2016]. Recent years have seen a surge in the development of multilabel ranking methods with strong practical and theoretical guarantees [Schapire and Singer, 2000, Dembczynski et al., 2012, Gong et al., 2013, Bucak et al., 2009, Jung and Tewari, 2018, Gao and Zhou, 2011, Koyejo et al., 2015, Zhang and Zhou, 2013, Korba et al., 2018]. A related line of work has studied consistency for the convex surrogates of natural ranking losses [Duchi et al., 2010, Buffoni et al., 2011, Gao and Zhou, 2011, Ravikumar et al., 2011, Calauzenes et al., 2012, Dembczynski et al., 2012]. Despite this vast literature on multilabel ranking, the fundamental question of when a multilabel ranking problem is *learnable* remains unanswered.

Understanding when a hypothesis class is learnable is a fundamental question in Statistical Learning Theory. For binary classification, the finiteness of the Vapnik–Chervonenkis (VC) dimension is both sufficient and necessary for Probably Approximately Correct (PAC) learning [Vapnik and Chervonenkis, 1974, Valiant, 1984]. Likewise, the finiteness of the Daniely-Shwartz (DS) dimension characterizes multiclass PAC learnability Daniely and Shalev-Shwartz [2014], Brukhim et al. [2022]. In the online setting, the Littlestone dimension [Littlestone, 1987] characterizes the online learnability of a binary hypothesis class and the multiclass Littlestone dimension [Daniely et al., 2011] characterizes online multiclass learnability. Unlike classification, a distinguishing property of multilabel ranking is the mismatch between the predictions the learner makes and the feedback it receives. In particular, a learner is required to produce a permutation that ranks the relevance of the labels but only receives a *relevance-score vector* as feedback. This feedback model is standard in multilabel ranking since obtaining full permutation feedback is generally costly [Liu et al., 2009]. As a result, unlike the 0-1 loss in classification, there is no canonical loss function in ranking. Together, these two issues

---

[*]Equal contributions

37th Conference on Neural Information Processing Systems (NeurIPS 2023).

create barriers for existing techniques used to prove learnability, such as the agnostic-to-realizable reductions from Hopkins et al. [2022] and Raman et al. [2023], to readily extend to ranking.

In this paper, we characterize the batch and online learnability of a ranking hypothesis class $\mathcal{H} \subseteq \mathcal{S}_K^{\mathcal{X}}$ under relevance-score feedback, where $\mathcal{S}_K$ is the set of all permutations over $[K] = \{1, ..., K\}$. In doing so, we make the following contributions.

- We show that a ranking hypothesis class $\mathcal{H}$ embeds $K^2$ different *binary* hypothesis classes $\mathcal{H}_i^j$ for $i, j \in [K]$, where hypotheses in $\mathcal{H}_i^j$ answer whether the label $i$ should be ranked in the top $j$. Our main result relates the learnability of $\mathcal{H}$ to the learnability of $\mathcal{H}_i^j$'s.
- We define two families of ranking loss functions that capture most if not all ranking losses used in practice. We show that these families are actually *equivalence* classes - the same characterization of batch and online learnability holds for every loss in that family.
- By relating the learnability of $\mathcal{H}$ to the learnability of *binary* hypothesis classes $\mathcal{H}_i^j$, we show that existing combinatorial dimensions, like the VC and Littlestone dimension, continue to characterize learnability in the multilabel ranking setting. This allows us to prove that linear ranking hypothesis classes are learnable in the batch setting.

A unifying theme throughout the paper is our ability to *constructively* convert a learning algorithm $\mathcal{A}$ for $\mathcal{H}$ into a learning algorithm $\mathcal{A}_i^j$ for $\mathcal{H}_i^j$ for each $i, j \in [K]$ and vice versa. To do so, our proof techniques involve adapting the agnostic-to-realizable reduction for batch and online classification, proposed by Hopkins et al. [2022] and Raman et al. [2023] respectively, to ranking.

## 2 Preliminaries and Notation

Let $\mathcal{X}$ denote the instance space, $\mathcal{S}_K$ the set of permutations over labels $[K] := \{1, ..., K\}$, and $\mathcal{Y} = \{0, 1, ..., B\}^K$ the target space for some $K, B \in \mathbb{N}$. We highlight that the set of labels $[K]$ is fixed beforehand and does not depend on the instance $x \in \mathcal{X}$. This is to be contrasted with subset ranking, the set of labels can change depending on the instance $x \in \mathcal{X}$.

We refer to an element $y \in \mathcal{Y}$ as a *relevance-score vector* that indicates the relevance of each of the $K$ labels, where $B$ indicates the highest relevance and 0 indicates the lowest relevance. Throughout the paper, we treat a permutation $\pi \in \mathcal{S}_K$ as a vector in $\{1, ..., K\}^K$ that induces a *ranking* of the $K$ labels in decreasing order of relevance. Accordingly, for an index $i \in [K]$, we let $\pi_i \in [K]$ denote the *rank* of label $i$. Likewise, given an index $i \in [K]$, we let $y^i$ denote the relevance of label $i$. In addition, it will be useful to define a mapping from $\mathcal{S}_K$ to $\{0, 1\}^K$. In particular, we define $\text{BinRel}(\cdot, \cdot) : \mathcal{S}_K \times [K] \to \{0, 1\}^K$ as an operator that given a permutation (ranking) $\pi \in \mathcal{S}_K$ and threshold $p \in [K]$, outputs a bit string $b \in \{0, 1\}^K$ s.t. $b_i = \mathbb{1}\{\pi_i \leq p\}$.

**Ranking Equivalences**. Our construction of ranking loss families in Section 3 requires different notions of equivalence between permutations (rankings) in $\mathcal{S}_K$. To that end, we say that $\pi = \hat{\pi}$ if and only if for all $i \in [K]$, $\pi_i = \hat{\pi}_i$. On the other hand, we say $\pi \overset{p}{=} \hat{\pi}$ if and only if $\{i : \pi_i \leq p\} = \{i : \hat{\pi}_i \leq p\}$. That is, two rankings are $p$-equivalent if the *set* of labels they rank in the top-$p$ are equal. Finally, we say $\pi \overset{[p]}{=} \hat{\pi}$ if and only if for all $j \in [p]$, $\{i : \pi_i \leq j\} = \{i : \hat{\pi}_i \leq j\}$. That is, two rankings are $[p]$-equivalent if not only the *set* but also the *order* of labels they rank in the top-$p$ are equal.

**Ranking Hypothesis**. A ranking hypothesis $h \in \mathcal{H} \subseteq \mathcal{S}_K^{\mathcal{X}}$ maps instances in $\mathcal{X}$ to a ranking (permutation) in $\mathcal{S}_K$. Given an instance $x \in \mathcal{X}$, one can think of $h(x)$ as $h$'s ranking of the $K$ different labels in decreasing order of relevance. For any ranking hypothesis $h$, we let $h_i : \mathcal{X} \to [K]$ denote its restriction to the $i$'th coordinate output. Accordingly, for an instance $x \in \mathcal{X}$, $h_i(x)$ gives the rank that $h$ assigns to label $i$. Given a ranking hypothesis class $\mathcal{H} \subseteq \mathcal{S}_K^{\mathcal{X}}$ and any $i, j \in [K]$, we define its binary threshold-restricted hypothesis class $\mathcal{H}_i^j = \{h_i^j : h \in \mathcal{H}\}$ where $h_i^j(x) = \mathbb{1}\{h_i(x) \leq j\}$. We can think of hypotheses in $\mathcal{H}_i^j$ as providing binary responses to queries of the form: "for instance $x$, should label $i$ ranked in the top $j$?" These threshold-restricted classes are central to our characterization of learnability in both the batch and online learning settings.

**Batch Learnability.** In the batch setting, we are interested in characterizing the learnability of a ranking hypothesis class $\mathcal{H}$ under a model similar to the classical PAC model [Valiant, 1984].

**Definition 1** (Agnostic Ranking PAC Learnability). *A ranking hypothesis class $\mathcal{H} \subseteq \mathcal{S}_K^{\mathcal{X}}$ is agnostic PAC learnable w.r.t. loss $\ell : \mathcal{S}_K \times \mathcal{Y} \to \mathbb{R}_{\geq 0}$, if there exists a function $m : (0,1)^2 \times \mathbb{N} \to \mathbb{N}$ and a learning algorithm $\mathcal{A} : (\mathcal{X} \times \mathcal{Y})^\star \to \mathcal{S}_K^{\mathcal{X}}$ with the following property: for every $\epsilon, \delta \in (0,1)$ and for every distribution $\mathcal{D}$ on $\mathcal{X} \times \mathcal{Y}$, running algorithm $\mathcal{A}$ on $n \geq m(\epsilon, \delta, K)$ iid samples from $\mathcal{D}$ outputs a predictor $g = \mathcal{A}(S)$ such that with probability at least $1 - \delta$ over $S \sim \mathcal{D}^n$,*

$$\mathbb{E}_{\mathcal{D}}[\ell(g(x), y)] \leq \inf_{h \in \mathcal{H}} \mathbb{E}_{\mathcal{D}}[\ell(h(x), y)] + \epsilon.$$

If $\mathcal{D}$ is restricted to the class of distributions such that $\inf_{h \in \mathcal{H}} \mathbb{E}_{\mathcal{D}}[\ell(h(x), y)] = 0$, then we say we are in the *realizable* setting. Note that unlike in classification, realizability in the multilabel ranking setting is loss dependent.

**Online Learnability.** In the online setting, an adversary plays a sequential game with the learner over $T$ rounds. In each round $t \in [T]$, an adversary selects a labeled instance $(x_t, y_t) \in \mathcal{X} \times \mathcal{Y}$ and reveals $x_t$ to the learner. The learner makes a (potentially randomized) prediction $\hat{\pi}_t \in \mathcal{S}_K$. Finally, the adversary reveals the true relevance-score vector $y_t$, and the learner suffers the loss $\ell(\hat{\pi}_t, y_t)$, where $\ell$ is some pre-specified ranking loss function. Given a ranking hypothesis class $\mathcal{H} \subseteq \mathcal{S}_K^{\mathcal{X}}$, the goal of the learner is to output predictions $\hat{\pi}_t$ such that its cumulative loss is close to the best possible cumulative loss over hypotheses in $\mathcal{H}$. A hypothesis class is online learnable if there exists an algorithm such that for any sequence of labeled examples $(x_1, y_1), ..., (x_T, y_T)$, the difference in cumulative loss between its predictions and the predictions of the best possible function in $\mathcal{H}$ is small.

**Definition 2** (Agnostic Online Ranking Learnability). *A ranking hypothesis class $\mathcal{H} \subseteq \mathcal{S}_K^{\mathcal{X}}$ is agnostic online learnable w.r.t. loss $\ell$, if there exists an (potentially randomized) algorithm $\mathcal{A}$ such that for any adaptively chosen sequence of labeled examples $(x_t, y_t) \in \mathcal{X} \times \mathcal{Y}$, the algorithm outputs $\mathcal{A}(x_t) \in \mathcal{S}_K$ at every iteration $t \in [T]$ such that its expected regret,*

$$R(T, K) := \mathbb{E}\left[\sum_{t=1}^{T} \ell(\mathcal{A}(x_t), y_t) - \inf_{h \in \mathcal{H}} \sum_{t=1}^{T} \ell(h(x_t), y_t)\right],$$

*is a non-decreasing, sub-linear function of $T$. Here, the expectation is taken with respect to the randomness of the algorithm $\mathcal{A}$.*

If it is further guaranteed that there exists a hypothesis $h^\star \in \mathcal{H}$ such that $\sum_{t=1}^{T} \ell(h^\star(x_t), y_t) = 0$, then we say we are in the *realizable* setting. Again, realizability is loss dependent.

## 3 Ranking Loss Families

In statistical learning theory, we often characterize learnability with respect to a loss function. Unlike the 0-1 loss in classification, there is no canonical loss function in multilabel ranking. Accordingly, we define two general families of ranking loss functions in this section and later characterize learnability with respect to all losses in these families. In Appendix A, we show that many of the ranking metrics used in practice (e.g. Pairwise Rank Loss, Discounted Cumulative Gain, Reciprocal Rank, Average Precision, Precision@p, etc.) fall into one of these two families.

On a high level, we can classify ranking losses into two main groups: (A) those losses that care about both the order and magnitude of the relevance scores within the top-$p$ ranked labels and (B) those losses that only care about the magnitude of the relevance scores within the top-$p$ ranked labels. Our goal will be to define a loss family for both groups A and B. To do so, we start by identifying a canonical ranking loss that lies in each group. For group A, the normalized sum loss@p,

$$\ell_{\text{sum}}^{@p}(\pi, y) = \sum_{i=1}^{K} \min(\pi_i, p+1) y^i - Z_y^p$$

captures both the order and magnitude of the relevance scores only for the top-$p$ ranked labels. Here, $Z_y^p$ is an appropriately chosen normalization factor that only depends on $p$ and $y$ such that

$\min_{\pi \in \mathcal{S}_K} \ell^{@p}_{\text{sum}}(\pi, y) = 0$. For Group B, the normalized precision loss@p,

$$\ell^{@p}_{\text{prec}}(\pi, y) = Z^p_y - \sum_{i=1}^K \mathbb{1}\{\pi_i \leq p\} y^i$$

cares only about the magnitude of relevance scores in the top-$p$ ranked labels. Again, $Z^p_y$ is an appropriately chosen normalization constant that only depends on $p$ and $y$ such that the minimum loss is 0. The form of $\ell^{@p}_{\text{prec}}$ differs from $\ell^{@p}_{\text{sum}}$ because $\sum_{i=1}^K \mathbb{1}\{\pi_i \leq p\} y^i$ is a gain whereas $\sum_{i=1}^K \min(\pi_i, p+1) y^i$ is a loss.

Next, we build loss families around $\ell^{@p}_{\text{sum}}$ and $\ell^{@p}_{\text{prec}}$. For $\ell^{@p}_{\text{sum}}$, consider the family:

$$\mathcal{L}(\ell^{@p}_{\text{sum}}) = \{\ell \in \mathbb{R}^{\mathcal{S}_K \times \mathcal{Y}} : \ell = 0 \text{ if and only if } \ell^{@p}_{\text{sum}} = 0\} \cap \{\ell \in \mathbb{R}^{\mathcal{S}_K \times \mathcal{Y}} : \pi \stackrel{[p]}{=} \hat{\pi} \implies \ell(\pi, y) = \ell(\hat{\pi}, y)\}.$$

By definition, $\mathcal{L}(\ell^{@p}_{\text{sum}})$ contains those ranking losses that are (1) zero-matched with $\ell^{@p}_{\text{sum}}$ and (2) remain unchanged for any two predicted rankings (permutations) that are $[p]$-equivalent. The second constraint is needed to ensure that losses in $\mathcal{L}(\ell^{@p}_{\text{sum}})$ only depend on the order and set of labels that $\pi$ ranks in the top-$p$. Likewise, we can construct a similar loss family around $\ell^{@p}_{\text{prec}}$ as follows:

$$\mathcal{L}(\ell^{@p}_{\text{prec}}) = \{\ell \in \mathbb{R}^{\mathcal{S}_K \times \mathcal{Y}} : \ell = 0 \text{ if and only if } \ell^{@p}_{\text{prec}} = 0\} \cap \{\ell \in \mathbb{R}^{\mathcal{S}_K \times \mathcal{Y}} : \pi \stackrel{p}{=} \hat{\pi} \implies \ell(\pi, y) = \ell(\hat{\pi}, y)\}.$$

The set $\mathcal{L}(\ell^{@p}_{\text{prec}})$ contains those ranking losses that are (1) zero-matched with $\ell^{@p}_{\text{prec}}$ and (2) remain unchanged for any two predicted rankings (permutations) that are $p$-*equivalent*. The second constraint is needed to ensure that losses in $\mathcal{L}(\ell^{@p}_{\text{prec}})$ only depend on the set of labels that $\pi$ ranks in the top-$p$. A major contribution of this paper is showing that both $\mathcal{L}(\ell^{@p}_{\text{sum}})$ and $\mathcal{L}(\ell^{@p}_{\text{prec}})$ are actually *equivalence* classes - the same characterization of learnability holds for every loss in that family.

# 4    Batch Multilabel Ranking

In this section, we characterize the agnostic PAC learnability of hypothesis classes $\mathcal{H} \subseteq \mathcal{S}_K^{\mathcal{X}}$ with respect to both $\mathcal{L}(\ell^{@p}_{\text{sum}})$ and $\mathcal{L}(\ell^{@p}_{\text{prec}})$. Our main results, stated below as two theorems, relate the learnability of $\mathcal{H}$ to the learnability of the threshold-restricted classes $\mathcal{H}_i^j$.

**Theorem 4.1.** *A hypothesis class $\mathcal{H} \subseteq \mathcal{S}_K^{\mathcal{X}}$ is agnostic PAC learnable w.r.t $\ell \in \mathcal{L}(\ell^{@p}_{sum})$ if and only if for all $i \in [K]$ and $j \in [p]$, $\mathcal{H}_i^j$ is agnostic PAC learnable w.r.t the 0-1 loss.*

**Theorem 4.2.** *A hypothesis class $\mathcal{H} \subseteq \mathcal{S}_K^{\mathcal{X}}$ is agnostic PAC learnable w.r.t $\ell \in \mathcal{L}(\ell^{@p}_{prec})$ if and only if for all $i \in [K]$, $\mathcal{H}_i^p$ is agnostic PAC learnable w.r.t the 0-1 loss.*

Since VC dimension characterizes the learnability of binary hypothesis classes under the 0-1 loss, an important corollary of Theorems 4.1 and 4.2 is that finiteness of VC($\mathcal{H}_i^j$)'s, for the appropriate $i, j \in [K] \times [p]$, is necessary and sufficient for agnostic ranking PAC learnability. Later on, we use this fact to prove that linear ranking hypothesis classes are agnostic ranking PAC learnable.

We start with the proof of Theorem 4.1, which follows in three steps. First, we show that if for all $(i, j) \in [K] \times [p]$, $\mathcal{H}_i^j$ is agnostic PAC learnable w.r.t 0-1 loss, then Empirical Risk Minimization (ERM) is an agnostic PAC learner for $\mathcal{H}$ w.r.t $\ell^{@p}_{\text{sum}}$. Next, we show that if $\mathcal{H}$ is agnostic PAC learnable w.r.t $\ell^{@p}_{\text{sum}}$, then $\mathcal{H}$ is agnostic PAC learnable w.r.t any loss $\ell \in \mathcal{L}(\ell^{@p}_{\text{sum}})$. Our proof of the latter uses the realizable to agnostic conversion from Hopkins et al. [2022]. Finally, we prove the necessity direction - if $\mathcal{H}$ is agnostic PAC learnable w.r.t an arbitrary $\ell \in \mathcal{L}(\ell^{@p}_{\text{sum}})$, then for all $(i, j) \in [K] \times [p]$, $\mathcal{H}_i^j$ is agnostic PAC learnable w.r.t 0-1 loss. The proof of necessity direction also uses the realizable to agnostic conversion from Hopkins et al. [2022]. The proof of Theorem 4.2 follows exactly the same way as Theorem 4.1 with some minor changes. Thus, we only focus on the proof of Theorem 4.1 in this section and defer all discussion of Theorem 4.2 to Appendix C.3.

We begin with Lemma 4.3, which asserts that if $\mathcal{H}_i^j$ is agnostic PAC learnable for all $(i, j) \in [K] \times [p]$, then ERM is an agnostic PAC learner for $\mathcal{H}$ w.r.t $\ell^{@p}_{\text{sum}}$.

**Lemma 4.3.** *If for all $i \in [K]$ and $j \in [p]$, $\mathcal{H}_i^j$ is agnostic PAC learnable w.r.t the 0-1 loss, then ERM is an agnostic PAC learner for $\mathcal{H} \subseteq \mathcal{S}_K^{\mathcal{X}}$ w.r.t $\ell_{\text{sum}}^{@p}$.*

The proof of Lemma 4.3 exploits the nice structure of $\ell_{\text{sum}}^{@p}$ by upperbounding the empirical Rademacher complexity of the loss class $\ell_{\text{sum}}^{@p} \circ \mathcal{H} = \{(x, y) \mapsto \ell_{\text{sum}}^{@p}(h(x), y) : h \in \mathcal{H})\}$ and showing that it vanishes as the sample size $n$ becomes large. Then, standard uniform convergence arguments outlined in Proposition C.1 imply that ERM is an agnostic PAC learner for $\mathcal{H}$ w.r.t $\ell_{\text{sum}}^{@p}$. The full proof is in Appendix C.

Since arbitrary losses in $\mathcal{L}(\ell_{\text{sum}}^{@p})$ may not have nice analytical forms, Lemma 4.4 relates the learnability of an arbitrary loss $\ell \in \mathcal{L}(\ell_{\text{sum}}^{@p})$ to the learnability of $\ell_{\text{sum}}^{@p}$.

**Lemma 4.4.** *If $\mathcal{H} \subseteq \mathcal{S}_K^{\mathcal{X}}$ is agnostic PAC learnable w.r.t $\ell_{\text{sum}}^{@p}$, then $\mathcal{H}$ is agnostic PAC learnable w.r.t any $\ell \in \mathcal{L}(\ell_{\text{sum}}^{@p})$.*

*Proof.* (of Lemma 4.4) Fix $\ell \in \mathcal{L}(\ell_{\text{sum}}^{@p})$. Let $a = \min_{\pi, y}\{\ell(\pi, y) \mid \ell(\pi, y) \neq 0\}$ and $b = \max_{\pi, y} \ell(\pi, y)$. We need to show that if $\mathcal{H}$ is agnostic PAC learnable w.r.t $\ell_{\text{sum}}^{@p}$, then $\mathcal{H}$ is agnostic PAC learnable w.r.t $\ell$. We will do so in two steps. First, we will show that if $\mathcal{A}$ is an agnostic PAC learner for $\ell_{\text{sum}}^{@p}$, then $\mathcal{A}$ is also a *realizable* PAC learner for $\ell$. Next, we will show how to convert a realizable PAC learner for $\ell$ into an agnostic PAC learner for $\ell$ in a black-box fashion. The composition of these two pieces yields an agnostic PAC learner for $\mathcal{H}$ w.r.t $\ell$.

**Realizable PAC learnability of $\mathcal{H}$ w.r.t $\ell$.** If $\mathcal{H}$ is agnostic PAC learnable w.r.t $\ell_{\text{sum}}^{@p}$, then there exists a learning algorithm $\mathcal{A}$ with sample complexity $m(\epsilon, \delta, K)$ s.t. for any distribution $\mathcal{D}$ over $\mathcal{X} \times \mathcal{Y}$, with probability $1 - \delta$ over a sample $S \sim \mathcal{D}^n$ of size $n \geq m(\epsilon, \delta, K)$, the output predictor $g = \mathcal{A}(S)$ achieves $\mathbb{E}_{\mathcal{D}}\left[\ell_{\text{sum}}^{@p}(g(x), y))\right] \leq \inf_{h \in \mathcal{H}} \mathbb{E}_{\mathcal{D}}\left[\ell_{\text{sum}}^{@p}(h(x), y))\right] + \epsilon$. In the realizable setting, we are further guaranteed that there exists a hypothesis $h^\star \in \mathcal{H}$ s.t. $\mathbb{E}_{\mathcal{D}}\left[\ell(h^\star(x), y))\right] = 0$. Since $\ell \in \mathcal{L}(\ell_{\text{sum}}^{@p})$, this also implies that $\mathbb{E}_{\mathcal{D}}\left[\ell_{\text{sum}}^{@p}(h^\star(x), y))\right] = 0$. Therefore, under realizability and the fact that $\ell \leq b\,\ell_{\text{sum}}^{@p}$, we have $\mathbb{E}_{\mathcal{D}}\left[\ell(g(x), y))\right] \leq b\epsilon$. This completes the first part of the proof as we have shown that $\mathcal{A}$ is also a realizable PAC learner for $\mathcal{H}$ w.r.t $\ell$ with sample complexity $m(\frac{\epsilon}{b}, \delta, K)$.

**Realizable-to-agnostic conversion.** Now, we show how to convert the realizable PAC learner $\mathcal{A}$ for $\ell$ into an agnostic PAC learner for $\ell$ in a black-box fashion. For this step, we will extend the agnostic-to-realizable reduction proposed by Hopkins et al. [2022] to the ranking setting by accommodating the mismatch between the range space of $\mathcal{H}$ and the label space $\mathcal{Y}$. In particular, we will show that Algorithm 1 below converts a realizable PAC learner for $\ell$ into an agnostic PAC learner for $\ell$. Note that although input $\mathcal{A}$ is a realizable learner, the distribution $\mathcal{D}$ may not be realizable.

---

**Algorithm 1** Agnostic PAC learner for $\mathcal{H}$ w.r.t. $\ell$

**Input:** Realizable PAC learner $\mathcal{A}$ for $\mathcal{H}$, unlabeled and labeled samples $S_U \sim \mathcal{D}_{\mathcal{X}}^n$ and $S_L \sim \mathcal{D}^m$

1 For each $h \in \mathcal{H}_{|S_U}$, construct a dataset

$$S_U^h = \{(x_1, \tilde{y}_1), ..., (x_n, \tilde{y}_n)\} \text{ s.t. } \tilde{y}_i \sim \text{Unif}\{\text{BinRel}(h(x_i), 1), ..., \text{BinRel}(h(x_i), p)\}$$

2 Run $\mathcal{A}$ over all datasets to get $C(S_U) := \left\{\mathcal{A}(S_U^h) \mid h \in \mathcal{H}_{|S_U}\right\}$
3 Return $\hat{g} \in C(S_U)$ with the lowest empirical error over $S_L$ w.r.t. $\ell$.

---

Let $h^\star = \arg\min_{h \in \mathcal{H}} \mathbb{E}_{\mathcal{D}}\left[\ell(h(x), y)\right]$ denote the optimal predictor in $\mathcal{H}$ w.r.t $\mathcal{D}$. Consider the sample $S_U^{h^\star}$ and let $g = \mathcal{A}(S_U^{h^\star})$. We can think of $g$ as the output of $\mathcal{A}$ run over an i.i.d sample $S$ drawn from $\mathcal{D}^\star$, a joint distribution over $\mathcal{X} \times \mathcal{Y}$ defined procedurally by first sampling $x \sim \mathcal{D}_{\mathcal{X}}$, then independently sampling $j \sim \text{Unif}([p])$, and finally outputting the labeled sample $(x, \text{BinRel}(h^\star(x), j))$. Note that $\mathcal{D}^\star$ is indeed a realizable distribution (realized by $h^\star$) w.r.t both $\ell$ and $\ell_{\text{sum}}^{@p}$. Recall that $m_{\mathcal{A}}(\frac{\epsilon}{b}, \delta, K)$ is the sample complexity of $\mathcal{A}$. Since $\mathcal{A}$ is a realizable learner for $\mathcal{H}$ w.r.t $\ell$, we have that for $n \geq m_{\mathcal{A}}(\frac{a\epsilon}{2b^2p}, \delta/2, K)$, with probability at least $1 - \frac{\delta}{2}$, $\mathbb{E}_{\mathcal{D}^\star}\left[\ell(g(x), y)\right] \leq \frac{a\epsilon}{2bp}$.

Next, by Lemma E.1, we have $\ell(g(x), y) \leq \ell(h^\star(x), y) + \frac{bp}{a}\mathbb{E}_{j\sim\text{Unif}([p])}\left[\ell(g(x), \text{BinRel}(h^\star(x), j))\right]$ pointwise. Taking expectations on both sides of the inequality gives

$$\mathbb{E}_\mathcal{D}\left[\ell(g(x), y)\right] \leq \mathbb{E}_\mathcal{D}\left[\ell(h^\star(x), y)\right] + \frac{bp}{a}\mathbb{E}_{x\sim\mathcal{D}}\left[\mathbb{E}_{j\sim\text{Unif}([p])}\left[\ell(g(x), \text{BinRel}(h^\star(x), j))\right]\right]$$

$$\leq \mathbb{E}_\mathcal{D}\left[\ell(h^\star(x), y)\right] + \frac{\epsilon}{2}.$$

The last inequality follows from the definition of $\mathcal{D}^\star$, namely $\mathbb{E}_{\mathcal{D}^\star}\left[\ell(g(x), y)\right] = \mathbb{E}_{x\sim\mathcal{D}_\mathcal{X}}\mathbb{E}_{j\sim\text{Unif}([p])}\left[\ell(g(x), \text{BinRel}(h^\star(x), j))\right]$. This shows that $C(S_U)$ contains a hypothesis $g$ that generalizes well with respect to $\mathcal{D}$. Now we want to show that the predictor $\hat{g}$ returned in step 4 also has good generalization. Crucially, observe that $C(S_U)$ is a finite hypothesis class with cardinality at most $2^{nK}$. By standard Chernoff and union bounds, with probability at least $1 - \delta/2$, the empirical risk of every hypothesis in $C(S_U)$ on a sample of size $\geq \frac{8}{\epsilon^2}\log\frac{4|C(S_U)|}{\delta}$ is at most $\epsilon/4$ away from its true error. So, if $m = |S_L| \geq \frac{8}{\epsilon^2}\log\frac{4|C(S_U)|}{\delta}$, then with probability at least $1 - \delta/2$,

$$\frac{1}{|S_L|}\sum_{(x,y)\in S_L}\ell(g(x), y) \leq \mathbb{E}_\mathcal{D}\left[\ell(g(x), y)\right] + \frac{\epsilon}{4} \leq \mathbb{E}_\mathcal{D}\left[\ell(h^\star(x), y)\right] + \frac{3\epsilon}{4}.$$

Since $\hat{g}$ is the ERM on $S_L$ over $C(S)$, its empirical risk can be at most $\mathbb{E}_\mathcal{D}\left[\ell(h^\star(x), y)\right] + \frac{3\epsilon}{4}$. Given that the population risk of $\hat{g}$ can be at most $\epsilon/4$ away from its empirical risk, we have that

$$\mathbb{E}_\mathcal{D}[\ell(\hat{g}(x), y)] \leq \mathbb{E}_\mathcal{D}\left[\ell(h^\star(x), y)\right] + \epsilon.$$

Applying union bounds, the entire process succeeds with probability $1 - \delta$. We can upper bound the sample complexity of Algorithm 1, denoted $n(\epsilon, \delta, K)$, as

$$n(\epsilon, \delta, K) \leq m_\mathcal{A}\left(\frac{a\epsilon}{2b^2 p}, \delta/2, K\right) + O\left(\frac{1}{\epsilon^2}\log\frac{|C(S_U)|}{\delta}\right)$$

$$\leq m_\mathcal{A}\left(\frac{a\epsilon}{2b^2 p}, \delta/2, K\right) + O\left(\frac{Km_\mathcal{A}(\frac{a\epsilon}{2b^2 p}, \delta/2, K) + \log\frac{1}{\delta}}{\epsilon^2}\right),$$

where we use $|C(S_U)| \leq 2^{Km_\mathcal{A}(\frac{a\epsilon}{2b^2 p}, \delta/2, K)}$. This shows that Algorithm 1 is an agnostic PAC learner for $\mathcal{H}$ w.r.t $\ell$. $\qquad\square$

Finally, Lemma 4.5 gives the necessity direction of Theorem 4.1.

**Lemma 4.5.** *If a hypothesis class $\mathcal{H} \subseteq \mathcal{S}_K^\mathcal{X}$ is agnostic PAC learnable w.r.t $\ell \in \mathcal{L}(\ell_{sum}^{@p})$, then $\mathcal{H}_i^j$ is agnostic PAC learnable w.r.t the 0-1 loss for all $(i, j) \in [K] \times [p]$.*

Like the sufficiency proofs, the proof of Lemma 4.5 is constructive. Given an agnostic PAC learner for $\mathcal{H}$ w.r.t $\ell$, we construct an agnostic PAC learner for $\mathcal{H}_i^j$ w.r.t 0-1 loss using a slight modification of Algorithm 1. We defer the full proof to Appendix C since the analysis is similar to that of Algorithm 1. Together, Lemmas 4.3, 4.4 and 4.5 imply Theorem 4.1.

We conclude this section by giving a concrete application of our characterization. Consider the class of ranking-hypotheses $\mathcal{H} = \{x \mapsto \text{argsort}(Wx) : W \in \mathbb{R}^{K\times d}\}$ that compute rankings by sorting scores, in descending order, obtained from a linear function of the input features. Lemma 4.6, whose proof is in Appendix B, computes the VC dimension of $\mathcal{H}_i^j$ for an arbitrary $i, j \in [K]$.

**Lemma 4.6.** *Let $\mathcal{H} = \{x \mapsto \text{argsort}(Wx) : W \in \mathbb{R}^{K\times d}\}$ be a linear ranking hypothesis class. Then for all $i, j \in [K]$, $VC(\mathcal{H}_i^j) = \tilde{O}(Kd)$, where $\tilde{O}$ hides logarithmic factors of $d$ and $K$.*

Combining Lemma 4.6 with Theorems 4.1 and 4.2 shows that linear ranking hypothesis classes are agnostic ranking PAC learnable w.r.t to all losses in $\mathcal{L}(\ell_{sum}^{@p}) \cup \mathcal{L}(\ell_{prec}^{@p})$. More generally, in Appendix B we give a dimension-based sufficient condition under which *generic* score-based ranking hypothesis classes are agnostic ranking PAC learnable.

# 5 Online Multilabel Ranking

We now move to the online setting and characterize the online learnability of hypothesis classes $\mathcal{H} \subseteq \mathcal{S}_K^{\mathcal{X}}$ with respect to both $\mathcal{L}(\ell_{\text{sum}}^{@p})$ and $\mathcal{L}(\ell_{\text{prec}}^{@p})$. As in the batch setting, our characterization relates the learnability of $\mathcal{H}$ to the learnability of the threshold-restricted classes $\mathcal{H}_i^j$.

**Theorem 5.1.** *A hypothesis class $\mathcal{H} \subseteq \mathcal{S}_K^{\mathcal{X}}$ is agnostic online learnable w.r.t $\ell \in \mathcal{L}(\ell_{sum}^{@p})$ if and only if for all $i \in [K]$ and $j \in [p]$, $\mathcal{H}_i^j$ is agnostic online learnable w.r.t the 0-1 loss.*

**Theorem 5.2.** *A hypothesis class $\mathcal{H} \subseteq \mathcal{S}_K^{\mathcal{X}}$ is agnostic online learnable w.r.t $\ell \in \mathcal{L}(\ell_{prec}^{@p})$ if and only if for all $i \in [K]$, $\mathcal{H}_i^p$ is agnostic online learnable w.r.t the 0-1 loss.*

Since the Littlestone dimension characterizes the online learnability of binary hypothesis classes under the 0-1 loss, an important corollary of Theorems 5.1 and 5.2 is is that finiteness of $\text{Ldim}(\mathcal{H}_i^j)$, for the appropriate $i, j \in [K] \times [p]$, is necessary and sufficient for agnostic online ranking learnability.

We now begin the proof of Theorem 5.1. Since the proof of Theorem 5.2 follows a similar trajectory, we defer all discussion of Theorem 5.2 to Appendix D.2. Unlike Theorem 4.1 in the batch setting, we prove the sufficiency and necessity directions of Theorem 5.1 directly. We chose this direct path because, unlike the batch setting, sequential Rademacher analysis does not yield a constructive algorithm [Rakhlin et al., 2015]. On the other hand, our proofs are constructive and use the celebrated Randomized Exponential Weights Algorithm (REWA) [Cesa-Bianchi and Lugosi, 2006]. Moreover, a key ingredient of our proof is the realizable to agnostic conversion from Raman et al. [2023].

*Proof.* (of sufficiency in Theorem 5.1) Fix $\ell \in \mathcal{L}(\ell_{\text{sum}}^{@p})$. Let $a = \min_{\pi,y}\{\ell(\pi, y) \mid \ell(\pi, y) \neq 0\}$ and $M = \max_{\pi,y} \ell(\pi, y)$. Given online learners for $\mathcal{H}_i^j$ for the 0-1 loss, our goal is to construct an online learner $\mathcal{Q}$ for $\mathcal{H}$ w.r.t $\ell$ that enjoys sub-linear regret in $T$. Our strategy will be to construct a set of experts $\mathcal{E}$ using the online learners for $\mathcal{H}_i^j$'s and run REWA using $\mathcal{E}$ and an appropriately scaled version of $\ell$. Our proof borrows ideas from the realizable-to-agnostic online conversion from Raman et al. [2023] and so we use the same notation whenever possible.

Let $(x_1, y_1), ..., (x_T, y_T) \in (\mathcal{X} \times \mathcal{Y})^T$ denote the stream of points to be observed by the online learner. We will assume an oblivious adversary and thus the stream is fixed before the game starts. A standard reduction (Chapter 4 in Cesa-Bianchi and Lugosi [2006]) allows us to convert oblivious regret bounds to adaptive regret bounds. Since $\mathcal{H}_i^j \subseteq \{0, 1\}^{\mathcal{X}}$ is online learnable w.r.t. 0-1 loss, we are guaranteed the existence of online learners $\mathcal{A}_i^j$ for $\mathcal{H}_i^j$.

**Constructing Experts**. For any bitstring $b \in \{0, 1\}^T$, let $\phi : \{t \in [T] : b_t = 1\} \to \mathcal{S}_K$ denote a function mapping time points where $b_t = 1$ to rankings (permutations). Let $\Phi_b = \mathcal{S}_K^{\{t \in [T]: b_t=1\}}$ denote all such functions $\phi$. For every $h \in \mathcal{H}$, there exists a $\phi_b^h \in \Phi_b$ such that for all $t \in \{t : b_t = 1\}$, $\phi_b^h(t) = h(x_t)$. Let $|b| = |\{t \in [T] : b_t = 1\}|$. For every $b \in \{0, 1\}^T$ and $\phi \in \Phi_b$, we will define an Expert $E_{b,\phi}$. Expert $E_{b,\phi}$, formally presented in Algorithm 2, uses $\mathcal{A}_i^j$'s to make predictions in each round. However, $E_{b,\phi}$ only updates the $\mathcal{A}_i^j$'s on those rounds where $b_t = 1$, using $\phi$ to compute a labeled instance. For every $b \in \{0, 1\}^T$, let $\mathcal{E}_b = \bigcup_{\phi \in \Phi_b}\{E_{b,\phi}\}$ denote the set of all Experts parameterized by functions $\phi \in \Phi_b$. If $b$ is the bitstring with all zeros, then $\mathcal{E}_b$ will be empty. Therefore, we will actually define $\mathcal{E}_b = \{E_0\} \cup \bigcup_{\phi \in \Phi_b}\{E_{b,\phi}\}$, where $E_0$ is the expert that never updates $\mathcal{A}_i^j$'s and only uses them for predictions in all $t \in [T]$. Note that $1 \leq |\mathcal{E}_b| \leq (K!)^{|b|} \leq K^{K|b|}$.

Using these experts, Algorithm 3 presents our agnostic online learner $\mathcal{Q}$ for $\mathcal{H}$ w.r.t $\ell \in \mathcal{L}(\ell_{\text{sum}}^{@p})$. We now show that $\mathcal{Q}$ enjoys sub-linear regret. We highlight that there are three sources of randomness in online learner $\mathcal{Q}$, namely the randomness of sampling $B$, the internal randomness of $\mathcal{A}_i^j$'s, and the internal randomness of $\mathcal{P}$. One may think of internal randomness as arising from the sampling step involved in the randomized predictions. Let $A$ be the random variable associated with joint internal randomness of $\mathcal{A}_i^j$ for all $(i, j) \in [K] \times [p]$. Similarly, denote $P$ to be the random variable associated with the internal randomness of $\mathcal{P}$. We begin by using the guarantee of REWA.

**Algorithm 2** Expert $(b, \phi)$

**Input:** Independent copy of realizable learners $\mathcal{A}_i^j$ of $\mathcal{H}_i^j$ for each $(i, j) \in [K] \times [p]$
1 **for** $t = 1, ..., T$ **do**
2      Receive example $x_t$
3      Define a binary vote matrix $V_t \in \{0, 1\}^{K \times p}$ such that $V_t[i, j] = \mathcal{A}_i^j(x_t)$
4      Predict $\hat{\pi}_t \in \arg\min_{\pi \in \mathcal{S}_K} \langle \pi, V_t \mathbf{1}_p \rangle$
5      **if** $b_t = 1$ **then**
6          Let $\pi = \phi(t)$ and for all $(i, j) \in [K] \times [p]$, update $\mathcal{A}_i^j$ by passing $(x_t, \pi_i^j)$
7 **end**

---

**Algorithm 3** Agnostic Online Learner $\mathcal{Q}$ for $\mathcal{H}$ w.r.t. $\ell$

**Input:** Parameter $0 < \beta < 1$
1 Let $B \in \{0, 1\}^T$ s.t. $B_t \overset{\text{iid}}{\sim} \text{Bernoulli}(\frac{T^\beta}{T})$
2 Construct the set of experts $\mathcal{E}_B = \{E_0\} \cup \bigcup_{\phi \in \Phi_B} \{E_{B,\phi}\}$ according to Algorithm 2
3 Run REWA $\mathcal{P}$ using $\mathcal{E}_B$ and the loss function $\frac{\ell}{M}$ over the stream $(x_1, y_1), ..., (x_T, y_T)$

---

**REWA Guarantee**. Using Theorem 21.11 in Shalev-Shwartz and Ben-David [2014] and the fact that $B, A$ and $P$ are mutually independent, REWA guarantees almost surely that

$$\sum_{t=1}^T \mathbb{E}\left[\ell(\mathcal{P}(x_t), y_t)|B, A\right] \leq \inf_{E \in \mathcal{E}_B} \sum_{t=1}^T \ell(E(x_t), y_t) + M\sqrt{2T \ln(|\mathcal{E}_B|)}.$$

Taking an outer expectation gives

$$\mathbb{E}\left[\sum_{t=1}^T \ell(\mathcal{P}(x_t), y_t)\right] \leq \mathbb{E}\left[\inf_{E \in \mathcal{E}_B} \sum_{t=1}^T \ell(E(x_t), y_t)\right] + \mathbb{E}\left[M\sqrt{2T \ln(|\mathcal{E}_B|)}\right].$$

Noting that $\mathcal{Q}(x_t) = \mathcal{P}(x_t)$, we obtain

$$\mathbb{E}\left[\sum_{t=1}^T \ell(\mathcal{Q}(x_t), y_t)\right] \leq \mathbb{E}\left[\inf_{E \in \mathcal{E}_B} \sum_{t=1}^T \ell(E(x_t), y_t)\right] + \mathbb{E}\left[M\sqrt{2T \ln(|\mathcal{E}_B|)}\right]$$

$$\leq \mathbb{E}\left[\sum_{t=1}^T \ell(E_{B,\phi_B^{h^\star}}(x_t), y_t)\right] + M\mathbb{E}\left[\sqrt{2T \ln(|\mathcal{E}_B|)}\right].$$

In the last step, we used the fact that for all $b \in \{0, 1\}^T$ and $h \in \mathcal{H}$, $E_{b,\phi_b^h} \in \mathcal{E}_b$. Here, $h^\star = \inf_{h \in \mathcal{H}} \sum_{t=1}^T \ell(h(x_t), y_t)$ is the optimal function in hindsight. First, note that $\ln(|\mathcal{E}_B|) \leq K|B| \ln(K)$. Using Jensen's inequality gives $\mathbb{E}\left[\sqrt{2T \ln(|\mathcal{E}_B|)}\right] \leq \sqrt{2T^{1+\beta} K \ln K}$. Thus,

$$\mathbb{E}\left[\sum_{t=1}^T \ell(\mathcal{Q}(x_t), y_t)\right] \leq \underbrace{\mathbb{E}\left[\sum_{t=1}^T \ell(E_{B,\phi_B^{h^\star}}(x_t), y_t)\right]}_{\text{(I)}} + M\sqrt{2T^{1+\beta} K \ln K}. \tag{1}$$

**Upperbounding (I).** It now suffices to upperbound $\mathbb{E}\left[\sum_{t=1}^T \ell(E_{B,\phi_B^{h^\star}}(x_t), y_t)\right]$. Recall that Lemma E.1 gives pointwise

$$\ell(E_{B,\phi_B^{h^\star}}(x_t), y_t) \leq \ell(h^\star(x_t), y_t) + \frac{pM}{a} \mathbb{E}_{j \sim \text{Unif}([p])}[\ell(E_{B,\phi_B^{h^\star}}(x_t), \text{BinRel}(h^\star(x_t), j))] \tag{2}$$

where $M = \max_{\pi,y} \ell(\pi, y)$ and $a = \min_{\pi,y}\{\ell(\pi, y) \mid \ell(\pi, y) \neq 0\}$. Note that, by definition of the constant $M$, we further get

$$\ell(E_{B,\phi_B^{h^\star}}(x_t), \text{BinRel}(h^\star(x_t), j)) \leq M \mathbb{1}\{\ell(E_{B,\phi_B^{h^\star}}(x_t), \text{BinRel}(h^\star(x_t), j)) > 0\}$$

$$= M \mathbb{1}\{\ell_{\text{sum}}^{@p}(E_{B,\phi_B^{h^\star}}(x_t), \text{BinRel}(h^\star(x_t), j)) > 0\},$$

where the equality follows from the fact that $\ell \in \mathcal{L}(\ell_{\text{sum}}^{@p})$.

In order to upperbound the indicator above, we need to introduce some more notations. Given the realizable online learner $\mathcal{A}_i^m$ for $(i,m) \in [K] \times [p]$, an instance $x \in \mathcal{X}$, and an ordered finite sequence of labeled examples $L \in (\mathcal{X} \times \{0,1\})^*$, let $\mathcal{A}_i^m(x|L)$ be the random variable denoting the prediction of $\mathcal{A}_i^m$ on the instance $x$ after running and updating on $L$. For any $b \in \{0,1\}^T$, $h \in \mathcal{H}$, and $t \in [T]$, let $L_{b_{<t}}^h(i,m) = \{(x_s, h_i^m(x_s)) : s < t \text{ and } b_s = 1\}$ denote the *subsequence* of the sequence of labeled instances $\{(x_s, h_i^m(x_s))\}_{s=1}^{t-1}$ where $b_s = 1$. Then, for any $j \in [p]$, we have

$$\mathbb{1}\{\ell_{\text{sum}}^{@p}(E_{B,\phi_B^{h^\star}}(x_t), \text{BinRel}(h^\star(x_t), j)) > 0\} \leq \sum_{i=1}^K \sum_{m=1}^p \mathbb{1}\{\mathcal{A}_i^m(x_t \mid L_{B_{<t}}^{h^\star}(i,m)) \neq h_i^{\star,m}(x_t)\}.$$

To prove the inequality above, consider the case when $\sum_{i=1}^K \sum_{m=1}^p \mathbb{1}\{\mathcal{A}_i^m(x_t \mid L_{B_{<t}}^{h^\star}(i,m)) \neq h_i^{\star,m}(x_t)\} = 0$ because the inequality is trivial otherwise. Then, we must have $\mathcal{A}_i^m(x_t \mid L_{B_{<t}}^{h^\star}(i,m)) = h_i^{\star,m}(x_t)$ for all $(i,m) \in [K] \times [p]$. Let $V_t \in \{0,1\}^{K \times p}$ be a binary vote matrix that $E_{B,\phi^{h^\star}}$ constructs in round $t$. Then, we have $V_t[i,m] = \mathcal{A}_i^m(x_t \mid L_{B_{<t}}^{h^\star}(i,m)) = h_i^{\star,m}(x_t)$ for all $(i,m) \in [K] \times [p]$. Since $h^\star(x_t)$ is a permutation, the vote vector $V_t \mathbf{1}_p$ must contain $p$ labels with distinct number of non-zero votes, namely $p, p-1, p-2, \ldots, 2, 1$ votes. Similarly, there must be $K - p$ labels with exactly 0 votes. Thus, every $\hat{\pi}_t \in \arg\min_{\pi \in \mathcal{S}_K} \langle \pi, V_t \mathbf{1}_p \rangle$ must rank label that obtained $p$ votes as 1, label with $p-1$ votes as 2, and so forth. In other words, we must have $\hat{\pi}_t \overset{[p]}{=} h^\star(x_t)$, and thus $\ell_{\text{sum}}^{@p}(\hat{\pi}_t, \text{BinRel}(h^\star(x_t), j)) = 0$ for any $j \in [p]$ by definition of $\ell_{\text{sum}}^{@p}$. Our claim now follows because $E_{B,\phi_B^{h^\star}}(x_t) \in \arg\min_{\pi \in \mathcal{S}_K} \langle \pi, V_t \mathbf{1}_p \rangle$. Using these two inequalities in equation (2), we obtain

$$\ell(E_{B,\phi_B^{h^\star}}(x_t), y_t) \leq \ell(h^\star(x_t), y_t) + \frac{pM^2}{a} \sum_{i=1}^K \sum_{m=1}^p \mathbb{1}\{\mathcal{A}_i^m(x_t \mid L_{B_{<t}}^{h^\star}(i,m)) \neq h_i^{\star,m}(x_t)\},$$

which further implies that

$$\mathbb{E}\left[\sum_{t=1}^T \ell(E_{B,\phi_B^{h^\star}}(x_t), y_t)\right] \leq \sum_{t=1}^T \ell(h^\star(x_t), y_t) + \frac{pM^2}{a} \sum_{i=1}^K \sum_{m=1}^p \underbrace{\mathbb{E}\left[\sum_{t=1}^T \mathbb{1}\{\mathcal{A}_i^m(x_t \mid L_{B_{<t}}^{h^\star}(i,m)) \neq h_i^{\star,m}(x_t)\}\right]}_{\text{(II)}}.$$

The first term above is the cumulative loss of the best-fixed hypothesis in hindsight.

**Upperbounding (II).** It now suffices to show that $\mathbb{E}\left[\sum_{t=1}^T \mathbb{1}\{\mathcal{A}_i^m(x_t \mid L_{B_{<t}}^{h^\star}(i,m)) \neq h_i^{\star,m}(x_t)\}\right]$ is sub-linear for every $(i,m) \in [K] \times [p]$. Note that we can write

$$\mathbb{E}\left[\sum_{t=1}^T \mathbb{1}\{\mathcal{A}_i^m(x_t \mid L_{B_{<t}}^{h^\star}(i,m)) \neq h_i^{\star,m}(x_t)\}\right] = \sum_{t=1}^T \mathbb{E}\left[\mathbb{1}\{\mathcal{A}_i^m(x_t \mid L_{B_{<t}}^{h^\star}(i,m)) \neq h_i^{\star,m}(x_t)\}\right] \frac{\mathbb{E}[\mathbb{1}\{B_t = 1\}]}{\mathbb{E}[\mathbb{1}\{B_t = 1\}]}$$

$$= \frac{T}{T^\beta} \sum_{t=1}^T \mathbb{E}\left[\mathbb{1}\{\mathcal{A}_i^m(x_t \mid L_{B_{<t}}^{h^\star}(i,m)) \neq h_i^{\star,m}(x_t)\} \mathbb{1}\{B_t = 1\}\right],$$

where the last equality follows because $\mathbb{E}[\mathbb{1}\{B_t = 1\}] = \frac{T^\beta}{T}$ and the prediction of $\mathcal{A}_i^m(x_t \mid L_{B_{<t}}^{h^\star}(i,m))$ on round $t$ only depends on bitstring $(B_1, \ldots, B_{t-1})$, but is independent of $B_t$. Next, we can use the regret guarantee of algorithm $\mathcal{A}_i^m$ on the rounds it was updated. That is,

$$\sum_{t=1}^T \mathbb{E}\left[\mathcal{A}_i^m(x_t \mid L_{B_{<t}}^{h^\star}(i,m)) \mathbb{1}\{B_t = 1\}\right] = \mathbb{E}\left[\sum_{t:B_t=1} \mathcal{A}_i^m(x_t \mid L_{B_{<t}}^{h^\star}(i,m)) \neq h_i^{\star,m}(x_t)\right]$$

$$= \mathbb{E}\left[\mathbb{E}\left[\sum_{t:B_t=1} \mathcal{A}_i^m(x_t \mid L_{B_{<t}}^{h^\star}(i,m)) \neq h_i^{\star,m}(x_t) \middle| B\right]\right] \leq \mathbb{E}[R_i^m(|B|)],$$

where $R_i^m(|B|)$ is the regret of $\mathcal{A}_i^m$, a sub-linear function of $|B|$. In the last step, we use the fact that $\mathcal{A}_i^m$ is a realizable algorithm for $\mathcal{H}_i^m$ and the feedback that the algorithm received was $(x_t, h_i^{\star,m}(x_t))$

in the rounds whenever $B_t = 1$. Next, Lemma 5.17 from Ceccherini-Silberstein et al. [2017] guarantees that there exists a concave sub-linear function $\tilde{R}_i^m(|B|)$ that upperbounds $R_i^m(|B|)$. Thus, by Jensen's inequality, $\mathbb{E}_B\left[R_i^m(|B|)\right] \leq \mathbb{E}_B\left[\tilde{R}_i^m(|B|)\right] \leq \tilde{R}_i^m(\mathbb{E}_B\left[|B|\right]) \leq \tilde{R}_i^m(T^\beta)$, a sub-linear function of $T^\beta$.

Combining (I) and (II) together, we obtain

$$\mathbb{E}\left[\sum_{t=1}^T \ell(\mathcal{Q}(x_t), y_t)\right] \leq \inf_{h \in \mathcal{H}} \sum_{t=1}^T \ell(h(x_t), y_t) + \frac{pM^2}{a} \sum_{i=1}^K \sum_{m=1}^p \frac{T}{T^\beta} \tilde{R}_i^m(T^\beta) + M\sqrt{2T^{1+\beta}K \ln K}.$$

Since $\tilde{R}_i^m(T^\beta)$ is a sublinear function of $T^\beta$, we have that $\frac{T}{T^\beta}\tilde{R}_i^m(T^\beta)$ is a sublinear function of $T$. As the sum of sublinear functions is sublinear, the second term above must be a sublinear function of $T$. The regret is sub-linear for any choice of $\beta \in (0, 1)$. This completes our proof as we have shown that the algorithm $\mathcal{Q}$ achieves sub-linear regret in $T$. $\qquad\square$

The proof of the necessity direction of Theorem 5.1 also involves constructing experts and running the REWA algorithm. Since the argument is similar, we defer details to Appendix D.1.

# 6 Discussion

In this paper, we characterize the learnability of a multilabel ranking hypothesis class in both the batch and online setting for a wide range of practical ranking losses. In all cases, we show that a ranking hypothesis class is learnable if and only if a sufficient number of its binary-valued threshold restrictions are learnable. Our paper studies two families of ranking loss functions and leaves it open to characterize the learnability of other natural ranking loss functions. One loss function not captured by our families is recall@p.

While we do establish quantitative bounds on the sample complexity and regret, our bounds are not optimal. It may be difficult to improve the sample complexity and regret bound at the highest level of generality for all losses in the families considered here. However, for natural losses such as sum loss, it is an interesting future direction to derive the optimal sample complexity and regret bounds in both the realizable and agnostic settings. In addition, our bounds depend on the number of labels $K$. Recently, $K$-free bounds have been achieved for multiclass classification problems in both batch and online settings [Brukhim et al., 2022, Hanneke et al., 2023]. An interesting future direction is to study whether $K$-free bounds are possible for multilabel ranking.

Finally, the focus of this paper is on characterizing learnability, and thus our algorithms are not computationally efficient. A natural future direction is to construct computationally efficient algorithms for multilabel ranking. Along this direction, since ERM is the most common algorithm used in practice, it is an important future direction to tightly quantify the sample complexity of ERM in the batch setting. Moreover, in learning theory, combinatorial dimensions play an important role in providing a tight quantitative characterization of learnability. Thus, it is an interesting future direction to identify combinatorial dimensions that characterize multilabel ranking learnability for specific loss functions.

## Acknowledgements

We acknowledge the support of NSF via grant IIS-2007055. VR acknowledges the support of the NSF Graduate Research Fellowship.

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
