# A  Categorizing Popular Ranking Losses

Table 1: Categorizing Popular Ranking Losses.

| Loss | Loss Family |
|---|---|
| Sum Loss@p | $\mathcal{L}(\ell_{\text{sum}}^{@p})$ |
| Precision Loss@p | $\mathcal{L}(\ell_{\text{prec}}^{@p})$ |
| Average Precision | $\mathcal{L}(\ell_{\text{sum}}^{@K})$ |
| Area Under the Curve | $\mathcal{L}(\ell_{\text{sum}}^{@K})$ |
| Reciprocal Rank | $\mathcal{L}(\ell_{\text{prec}}^{@1})$ |
| Pairwise Rank Loss | $\mathcal{L}(\ell_{\text{sum}}^{@K})$ |
| Discounted Cumulative Loss | $\mathcal{L}(\ell_{\text{sum}}^{@K})$ |
| Discounted Cumulative Loss@p | $\mathcal{L}(\ell_{\text{sum}}^{@p})$ |

In this section, we show that our loss families $\mathcal{L}(\ell_{\text{sum}}^{@p})$ and $\mathcal{L}(\ell_{\text{prec}}^{@p})$ are general and capture many of the popular ranking loss functions used in practice. We summarize the results in Table 1.

Recall that

$$\mathcal{L}(\ell_{\text{sum}}^{@p}) = \{\ell \in \mathbb{R}^{\mathcal{S}_K \times \mathcal{Y}} : \ell = 0 \text{ if and only if } \ell_{\text{sum}}^{@p} = 0\} \cap \{\ell \in \mathbb{R}^{\mathcal{S}_K \times \mathcal{Y}} : \pi \overset{[p]}{=} \hat{\pi} \implies \ell(\pi, y) = \ell(\hat{\pi}, y)\},$$

where

$$\ell_{\text{sum}}^{@p}(\pi, y) = \sum_{i=1}^{K} \min(\pi_i, p+1) y^i - Z_y^p.$$

Note that the normalization constant is defined as $Z_y^p := \min_{\pi \in \mathcal{S}_K} \sum_{i=1}^{K} \min(\pi_i, p+1) y^i$ and thus only depends on $y$. Furthermore,

$$\mathcal{L}(\ell_{\text{prec}}^{@p}) = \{\ell \in \mathbb{R}^{\mathcal{S}_K \times \mathcal{Y}} : \ell = 0 \text{ if and only if } \ell_{\text{prec}}^{@p} = 0\} \cap \{\ell \in \mathbb{R}^{\mathcal{S}_K \times \mathcal{Y}} : \pi \overset{p}{=} \hat{\pi} \implies \ell(\pi, y) = \ell(\hat{\pi}, y)\}.$$

where

$$\ell_{\text{prec}}^{@p}(\pi, y) = Z_y^p - \sum_{i=1}^{K} \mathbb{1}\{\pi_i \leq p\} y^i.$$

As before, the normalization constant $Z_y^p := \max_{\pi \in \mathcal{S}_K} \sum_{i=1}^{K} \mathbb{1}\{\pi_i \leq p\} y^i$ only depends on $y$.

In ranking literature, many evaluation metrics are often stated in terms of *gain* functions. However, these can be easily converted into loss functions by subtracting the gain from the maximum possible value of the gain. When relevance scores are restricted to be binary (i.e. $\mathcal{Y} = \{0, 1\}^K$), the **Average Precision** (AP) metric is a *gain* function defined as

$$\text{AP}(\pi, y) = \frac{1}{\|y\|_1} \sum_{i \in \{\pi_m : y^m = 1\}} \frac{\sum_{j=1}^{K} \mathbb{1}\{\pi_j \leq i\} y^j}{i}.$$

Since the maximum value AP can take is 1, we can define its loss function variant as:

$$\ell_{\text{AP}}(\pi, y) = 1 - \text{AP}(\pi, y).$$

Note that $\ell_{\text{AP}}(\pi, y) = 0$ if and only if $\pi$ ranks all labels where $y_i = 1$ in the top $\|y\|_1$. Therefore, $\ell_{\text{AP}}(\pi, y) \in \mathcal{L}(\ell_{\text{sum}}^{@K})$.

Another useful metric for binary relevance feedback is the **Area Under the Curve** (AUC) loss function:

$$\ell_{\text{AUC}}(\pi, y) = \frac{1}{\|y\|_1 (K - \|y\|_1)} \sum_{i=1}^{K} \sum_{j=1}^{K} \mathbb{1}\{\pi_i < \pi_j\} \mathbb{1}\{y^i < y^j\}.$$

The AUC computes the fraction of "bad pairs" of labels (i.e those pairs of labels where $i$ was more relevant than $j$, but $i$ was ranked lower than $j$). Again, note that $\ell_{\text{AUC}}(\pi, y) = 0$ if and only if $\pi$ ranks all labels where $y^i = 1$ in the top $\|y\|_1$. Therefore, $\ell_{\text{AP}}(\pi, y) \in \mathcal{L}(\ell_{\text{sum}}^{@K})$.

Lastly, the **Reciprocal Rank** (RR) metric is another important *gain* function for binary relevance score feedback,

$$\text{RR}(\pi, y) = \frac{1}{\min_{i:y^i=1} \pi_i}.$$

Its loss equivalent can be written as:

$$\ell_{\text{RR}}(\pi, y) = 1 - \text{RR}(\pi, y).$$

Since $\ell_{\text{RR}}(\pi, y)$ only cares about the relevance of the top-ranked label, we have that $\ell_{\text{RR}}(\pi, y) \in \mathcal{L}(\ell_{\text{prec}}^{@1})$.

Moving onto non-binary relevance scores, we start with the **Pairwise Rank Loss** (PL):

$$\ell_{\text{PL}}(\pi, y) = \sum_{i=1}^{K} \sum_{j=1}^{K} \mathbb{1}\{\pi_i < \pi_j\} \mathbb{1}\{y^i < y^j\}.$$

The Pairwise Ranking loss is the analog of AUC for non-binary relevance scores and thus $\ell_{\text{PL}}(\pi, y) \in \mathcal{L}(\ell_{\text{sum}}^{@K})$.

Finally, we have the **Discounted Cumulative Gain** (DCG) metric, defined as:

$$\text{DCG}(\pi, y) = \sum_{i=1}^{K} \frac{2^{y^i} - 1}{\log_2(1 + \pi_i)}.$$

For an appropriately chosen normalizing constant $Z_y$, we can define its associated loss:

$$\ell_{\text{DCG}}(\pi, y) = Z_y - \text{DCG}(\pi, y).$$

Like $\ell_{\text{sum}}^{@K}$, $\ell_{\text{DCG}}(\pi, y)$ is $0$ if and only if $\pi$ ranks the $K$ labels in increasing order of relevance, breaking ties arbitrarily. Thus, $\ell_{\text{DCG}}(\pi, y) \in \mathcal{L}(\ell_{\text{sum}}^{@K})$. If one only cares about the top-$p$ ranked results, then the DCG@p loss function evaluates only the top-$p$ ranked labels:

$$\ell_{\text{DCG}}^{@p}(\pi, y) = Z_y^p - \sum_{i=1}^{K} \frac{2^{y^i} - 1}{\log_2(1 + \pi_i)} \mathbb{1}\{\pi_i \le p\} = Z_y^p - \text{DCG}^{@p}(\pi, y).$$

Analogously, we have that $\ell_{\text{DCG}}^{@p}(\pi, y) \in \mathcal{L}(\ell_{\text{sum}}^{@p})$.

# B    Agnostic PAC Learnability of Score-based Rankers

In this section, we apply our results in the main paper to give sufficient conditions for the agnostic PAC learnability of score-based ranking hypothesis classes. A score-based ranking hypothesis $h : \mathcal{X} \to \mathcal{S}_K$ first maps an input $x \in \mathcal{X}$ to a vector in $\mathbb{R}^K$ representing the "score" for each label. Then, it outputs a ranking (permutation) over the labels in $[K]$ by sorting the real-valued vector in decreasing order of score.

More formally, let $\mathcal{F} \subseteq (\mathbb{R}^K)^{\mathcal{X}}$ denote a set of functions mapping elements from the input space $\mathcal{X}$ to score-vectors in $\mathbb{R}^K$. For each $f \in \mathcal{F}$, define the score-based ranking hypothesis $h_f(x) = \text{argsort}(f(x))$ which first computes the score-vector $f(x) \in \mathbb{R}^K$, and then outputs a ranking by sorting $f(x)$ in decreasing order, breaking ties by giving the smaller label the higher rank. That is, if $f_1(x) = f_2(x)$, then label 1 will be ranked higher than label 2. Given $\mathcal{F}$, define its induced score-based ranking hypothesis class as $\mathcal{H} = \{h_f : f \in \mathcal{F}\}$. Since our characterization of ranking learnability relates the learnability of $\mathcal{H}$ to the learnability of the *binary* threshold-restricted classes $\mathcal{H}_i^j = \{h_i^j : h \in \mathcal{H}\}$, it suffices to consider an arbitrary threshold-restricted class $\mathcal{H}_i^j$ and bound its VC dimension. Before we do so, we need some more notation regarding $\mathcal{F}$.

For each $k \in [K]$, define the scalar-valued function class $\mathcal{F}_k = \{f_k \mid (f_1, \ldots, f_K) \in \mathcal{F}\}$ by restricting each function in $\mathcal{F}$ to its $k^{th}$ coordinate output. Here, each $\mathcal{F}_k \subseteq \mathbb{R}^{\mathcal{X}}$ and we can write

$\mathcal{F} = (\mathcal{F}_1, \ldots, \mathcal{F}_K)$. For a function $f \in \mathcal{F}$, we will use $f_k(x)$ to denote the $k^{th}$ coordinate output of $f(x)$. For every $(i, j) \in [K] \times [K]$, define the function class $\mathcal{F}_i - \mathcal{F}_j = \{f_i - f_j : f \in \mathcal{F}\}$ where we let $f_i - f_j : \mathcal{X} \to \mathbb{R}$ denote a function such that $(f_i - f_j)(x) = f_i(x) - f_j(x)$. Subsequently, for any $(i, j) \in [K] \times [K]$, define the *binary* hypothesis classes $\mathcal{G}_{i,j} = \{\mathbb{1}\{(f_i - f_j)(x) < 0\} : f_i - f_j \in \mathcal{F}_i - \mathcal{F}_j\}$ and $\tilde{\mathcal{G}}_{i,j} = \{\mathbb{1}\{(f_i - f_j)(x) \leq 0\} : f_i - f_j \in \mathcal{F}_i - \mathcal{F}_j\}$. Finally, let $C_j : \{0, 1\}^K \to \{0, 1\}$ be the $K$-wise composition s.t. $C_j(b) = \mathbb{1}\{\sum_{i=1}^K b_i \leq j\}$ and define $C_j(\mathcal{G}_1, \ldots, \mathcal{G}_K) = \{C_j(g_1, \ldots, g_K) : (g_1, \ldots, g_K) \in \mathcal{G}_1 \times \ldots \times \mathcal{G}_K\}$. In other words, $C_j(\mathcal{G}_1, \ldots, \mathcal{G}_K)$ is the *binary* hypothesis class constructed by taking all combinations of binary classifiers from $\mathcal{G}_1, \ldots, \mathcal{G}_K$, summing them up, and thresholding the sum at $j$. We are now ready to bound the VC dimension of an arbitrary threshold-restricted class $\mathcal{H}_i^j$.

Consider an arbitrary threshold-restricted class $\mathcal{H}_i^j$ and hypothesis $h \in \mathcal{H}$. By definition, $h_i^j \in \mathcal{H}_i^j$. Let $f \in \mathcal{F}$ denote the function associated with $h$. Given an instance $x \in \mathcal{X}$, recall that $h_i^j(x) = \mathbb{1}\{h_i(x) \leq j\}$ where $h_i(x)$ is the rank that $h$ gives to the label $i$ for instance $x$. Since $h(x) = \text{argsort}(f(x))$, we have

$$h_i(x) = \text{argsort}(f(x))[i]$$
$$= \sum_{m=1}^{i} \mathbb{1}\{f_i(x) \leq f_m(x)\} + \sum_{m=i+1}^{K} \mathbb{1}\{f_i(x) < f_m(x)\}$$
$$= \sum_{m=1}^{i} \mathbb{1}\{(f_i - f_m)(x) \leq 0\} + \sum_{m=i+1}^{K} \mathbb{1}\{(f_i - f_m)(x) < 0\}$$

Thus, we can write:

$$h_i^j(x) = \mathbb{1}\left\{\left(\sum_{m=1}^{i} \mathbb{1}\{(f_i - f_m)(x) \leq 0\} + \sum_{m=i+1}^{K} \mathbb{1}\{(f_i - f_m)(x) < 0\}\right) \leq j\right\}.$$

Note that $h_i^j \in C_j(\tilde{\mathcal{G}}_{i,1}, \ldots, \tilde{\mathcal{G}}_{i,i}, \mathcal{G}_{i,i+1}, \ldots, \mathcal{G}_{i,K})$ by construction. Since $h$, and therefore $h_i^j$, was arbitrary, it further follows that $\mathcal{H}_i^j \subseteq C_j(\tilde{\mathcal{G}}_{i,1}, \ldots, \tilde{\mathcal{G}}_{i,i}, \mathcal{G}_{i,i+1}, \ldots, \mathcal{G}_{i,K})$. Therefore,

$$\text{VC}(\mathcal{H}_i^j) \leq \text{VC}(C_j(\tilde{\mathcal{G}}_{i,1}, \ldots, \tilde{\mathcal{G}}_{i,i}, \mathcal{G}_{i,i+1}, \ldots, \mathcal{G}_{i,K})).$$

Since $C_j(\tilde{\mathcal{G}}_{i,1}, \ldots, \tilde{\mathcal{G}}_{i,i}, \mathcal{G}_{i,i+1}, \ldots, \mathcal{G}_{i,K})$ is some $K$-wise composition of binary classes $\tilde{\mathcal{G}}_{i,1}, \ldots, \tilde{\mathcal{G}}_{i,i}, \mathcal{G}_{i,i+1}, \ldots, \mathcal{G}_{i,K}$, standard VC composition guarantees that $\text{VC}(C_j(\tilde{\mathcal{G}}_{i,1}, \ldots, \tilde{\mathcal{G}}_{i,i}, \mathcal{G}_{i,i+1}, \ldots, \mathcal{G}_{i,K})) = \tilde{O}(\text{VC}(\tilde{\mathcal{G}}_{i,1}) + \ldots + \text{VC}(\tilde{\mathcal{G}}_{i,i}) + \text{VC}(\mathcal{G}_{i,i+1}) + \ldots + \text{VC}(\mathcal{G}_{i,K}))$, where we hide log factors of $K$ and the VC dimensions [Dudley, 1978, Alon et al., 2020]. Putting things together, we have that

$$\text{VC}(\mathcal{H}_i^j) \leq \tilde{O}(\text{VC}(\tilde{\mathcal{G}}_{i,1}) + \ldots + \text{VC}(\tilde{\mathcal{G}}_{i,i}) + \text{VC}(\mathcal{G}_{i,i+1}) + \ldots + \text{VC}(\mathcal{G}_{i,K})).$$

An identical analysis can also be used to give sufficient conditions for the *online* learnability of score-based rankers in terms of the Littlestone dimensions of $\mathcal{H}_j^i$.

Now, we consider the special class of *linear* score-based ranker and prove Lemma 4.6.

*Proof.* (of Lemma 4.6) Let $\mathcal{X} = \mathbb{R}^d$ and $\mathcal{F} = \{f_W : W \in \mathbb{R}^{K \times d}\}$ s.t. $f_W(x) = Wx$. Consider the class of linear score-based rankers $\mathcal{H} = \{h_{f_W} : f_W \in \mathcal{F}\}$ where $h_{f_W}(x) = \text{argsort}(f_W(x)) = \text{argsort}(Wx)$ breaking ties in the same way mentioned above. Note for all $i \in [K]$, $\mathcal{F}_i = \{f_w : w \in \mathbb{R}^d\}$ where $f_w(x) = w^T x$. Furthermore, $\mathcal{F}_i - \mathcal{F}_j = \mathcal{F}_i = \mathcal{F}_j$. Therefore, for any $(i, j) \in [K] \times [K]$,

$$\mathcal{G}_{i,j} = \{\mathbb{1}\{(f_i - f_j)(x) < 0\} : f_i - f_j \in \mathcal{F}_i - \mathcal{F}_j\} = \{\mathbb{1}\{f_w(x) < 0\} : w \in \mathbb{R}^d\}$$

and

$$\tilde{\mathcal{G}}_{i,j} = \{\mathbb{1}\{(f_i - f_j)(x) \leq 0\} : f_i - f_j \in \mathcal{F}_i - \mathcal{F}_j\} = \{\mathbb{1}\{f_w(x) \leq 0\} : w \in \mathbb{R}^d\}$$

are the set of half-space classifiers passing through the origin with dimension $d$. Since for all $(i, j) \in [K] \times [K]$, $\text{VC}(\tilde{\mathcal{G}}_{i,j}) = \text{VC}(\mathcal{G}_{i,j}) = d$, we get that $\text{VC}(\mathcal{H}_i^j) \leq \tilde{O}(Kd)$. $\qquad\square$

## C   Proofs for Batch Multilabel Ranking

Since many of the ranking losses we consider map to values in $\mathbb{R}$, the *empirical* Rademacher complexity will be a useful tool for proving learnability in the batch setting.

**Definition 3** (Empirical Rademacher Complexity of Loss Class)**.** *Let $\ell(\cdot, \cdot)$ be a loss function, $S = \{(x_1, y_1), ..., (x_n, y_n)\} \in (\mathcal{X} \times \mathcal{Y})^*$ be a set of examples, and $\ell \circ \mathcal{H} = \{(x, y) \mapsto \ell(h(x), y) : h \in \mathcal{H}\}$ be a loss class. The empirical Rademacher complexity of $\ell \circ \mathcal{H}$ is defined as*

$$\hat{\mathfrak{R}}_n(\ell \circ \mathcal{H}) = \mathbb{E}_\sigma \left[ \sup_{h \in \mathcal{H}} \left( \frac{1}{n} \sum_{i=1}^n \sigma_i \ell(h(x_i), y_i) \right) \right]$$

*where $\sigma_1, ..., \sigma_n$ are independent* Rademacher *random variables.*

In particular, a standard result relates the empirical Rademacher complexity to the generalization error of hypotheses in $\mathcal{H}$ with respect to a real-valued bounded loss function $\ell(h(x), y)$ [Bartlett and Mendelson, 2002].

**Proposition C.1** (Rademacher-based Uniform Convergence)**.** *Let $\mathcal{D}$ be a distribution over $\mathcal{X} \times \mathcal{Y}$ and $\ell(\cdot, \cdot) \leq c$ be a bounded loss function. With probability at least $1 - \delta$ over the sample $S \sim \mathcal{D}^n$, for all $h \in \mathcal{H}$ simultaneously,*

$$\left| \mathbb{E}_\mathcal{D}[\ell(h(x), y)] - \hat{\mathbb{E}}_S[\ell(h(x), y)] \right| \leq 2\hat{\mathfrak{R}}_n(\mathcal{F}) + O\left( c\sqrt{\frac{\ln(\frac{1}{\delta})}{n}} \right)$$

*where $\hat{\mathbb{E}}_S[\ell(h(x), y)] = \frac{1}{|S|} \sum_{(x,y) \in S} \ell(h(x), y)$ is the empirical average of the loss over $S$.*

When the empirical Rademacher complexity of the loss class $\ell \circ \mathcal{H} = \{(x, y) \mapsto \ell(h(x), y) : h \in \mathcal{H}\}$ is $o(1)$, we state that $\mathcal{H}$ enjoys the uniform convergence property w.r.t $\ell$. If $\mathcal{H}$ enjoys the uniform convergence property w.r.t. a loss $\ell$, a standard result shows that $\mathcal{H}$ is learnable according to Definition 1 via Empirical Risk Minimization (ERM) (Theorem 26.5 in Shalev-Shwartz and Ben-David [2014]).

### C.1   Proof of Lemma 4.3

*Proof.* Let $\mathcal{H} \subseteq \mathcal{S}_K^\mathcal{X}$ be an arbitrary ranking hypothesis class. We need to show that if $\mathcal{H}_i^j$ is agnostic PAC learnable w.r.t to 0-1 loss for all $(i, j) \in [K] \times [p]$, then ERM is an agnostic PAC learnable w.r.t $\ell_{\text{sum}}^{@p}$. By Proposition C.1, it suffices to show that the empirical Rademacher complexity of the loss class $\ell_{\text{sum}}^{@p} \circ \mathcal{H}$ vanishes as $n$ increases. This will imply that $\ell_{\text{sum}}^{@p}$ enjoys the uniform convergence property, and therefore ERM is an agnostic PAC learner for $\mathcal{H}$ w.r.t $\ell_{\text{sum}}^{@p}$. By definition, we have that

$$\hat{\mathfrak{R}}_n(\ell_{\text{sum}}^{@p} \circ \mathcal{H}) = \mathbb{E}_{\sigma \sim \{\pm 1\}^n}\left[\sup_{h \in \mathcal{H}} \frac{1}{n}\sum_{i=1}^{n} \sigma_i \ell_{\text{sum}}^{@p}(h(x_i), y_i))\right]$$

$$= \mathbb{E}_{\sigma \sim \{\pm 1\}^n}\left[\sup_{h \in \mathcal{H}} \frac{1}{n}\sum_{i=1}^{n}\left(\sum_{m=1}^{K} \sigma_i \min(h_m(x_i), p+1)y_i^m - \sigma_i Z_{y_i}^p\right)\right]$$

$$= \mathbb{E}_{\sigma \sim \{\pm 1\}^n}\left[\sup_{h \in \mathcal{H}} \frac{1}{n}\sum_{i=1}^{n}\sum_{m=1}^{K} \sigma_i \min(h_m(x_i), p+1)y_i^m\right]$$

$$\leq \sum_{m=1}^{K}\mathbb{E}_{\sigma \sim \{\pm 1\}^n}\left[\sup_{h \in \mathcal{H}} \frac{1}{n}\sum_{i=1}^{n} \sigma_i \min(h_m(x_i), p+1)y_i^m\right]$$

$$\leq B\sum_{m=1}^{K}\mathbb{E}_{\sigma \sim \{\pm 1\}^n}\left[\sup_{h \in \mathcal{H}} \frac{1}{n}\sum_{i=1}^{n} \sigma_i \min(h_m(x_i), p+1)\right]$$

where the second inequality follows from the fact that $y_i^m \leq B$ and Talagrand's Contraction Lemma Ledoux and Talagrand [1991].

Next note that $\min(h_m(x_i), p+1) = (p+1) - \sum_{j=1}^{p} \mathbb{1}\{h_m(x_i) \leq j\} = (p+1) - \sum_{j=1}^{p} h_m^j(x_i)$. Substituting and getting rid of constant factors, we have that

$$\hat{\mathfrak{R}}_n(\ell_{\text{sum}}^{@p} \circ \mathcal{H}) \leq B\sum_{m=1}^{K}\mathbb{E}_{\sigma \sim \{\pm 1\}^n}\left[\sup_{h_m \in \mathcal{H}_m} \frac{1}{n}\sum_{i=1}^{n} \sigma_i \sum_{j=1}^{p} h_m^j(x_i)\right]$$

$$\leq B\sum_{m=1}^{K}\sum_{j=1}^{p}\mathbb{E}_{\sigma \sim \{\pm 1\}^n}\left[\sup_{h_m \in \mathcal{H}_m} \frac{1}{n}\sum_{i=1}^{n} \sigma_i h_m^j(x_i)\right]$$

$$= B\sum_{m=1}^{K}\sum_{j=1}^{p}\hat{\mathfrak{R}}_n(\mathcal{H}_m^j).$$

Since for $\mathcal{H}_m^j$ is agnostic PAC learnable w.r.t 0-1 loss, by Theorem 6.5 in Shalev-Shwartz and Ben-David [2014], $\lim_{n \to \infty} \hat{\mathfrak{R}}_n(\mathcal{H}_m^j) = 0$. Since $p$, $K$ and $B$ are finite,

$$\lim_{n \to \infty} \hat{\mathfrak{R}}_n(\ell_{\text{sum}}^{@p} \circ \mathcal{H}) = \lim_{n \to \infty} B\sum_{m=1}^{K}\sum_{j=1}^{p}\hat{\mathfrak{R}}_n(\mathcal{H}_m^j) = 0$$

.

By Proposition C.1, this implies that $\ell_{\text{sum}}^{@p}$ enjoys the uniform convergence property, and therefore ERM using $\ell_{\text{sum}}^{@p}$ is an agnostic PAC learner for $\mathcal{H}$. $\qquad\square$

### C.2 Proof of Lemma 4.5

*Proof.* Fix $\ell \in \mathcal{L}(\ell_{\text{sum}}^{@p})$ and $(i,j) \in [K] \times [p]$. Let $a = \min_{\pi,y}\{\ell(\pi, y) \mid \ell(\pi, y) \neq 0\}$. Let $\mathcal{H}$ be an arbitrary ranking hypothesis class and $\mathcal{A}$ be an agnostic PAC learner for $\mathcal{H}$ w.r.t $\ell$. Our goal will be to use $\mathcal{A}$ to construct an agnostic PAC learner for $\mathcal{H}_i^j$.

Let $\mathcal{D}$ be distribution over $\mathcal{X} \times \{0, 1\}$ and $h_i^{\star,j} = \arg\min_{h_i^j \in \mathcal{H}_i^j} \mathbb{E}_{\mathcal{D}}\left[\mathbb{1}\{h_i^j(x) \neq y\}\right]$ be the optimal hypothesis. Let $h^\star \in \mathcal{H}$ be any valid completion of $h_i^{\star,j}$. Our goal will be to show that Algorithm 4 is an agnostic PAC learner for $\mathcal{H}_i^j$ w.r.t 0-1 loss.

Consider the sample $S_U^{h^\star}$ and let $g = \mathcal{A}(S_U^{h^\star})$. We can think of $g$ as the output of $\mathcal{A}$ run over an i.i.d sample $S$ drawn from $\mathcal{D}^\star$, a joint distribution over $\mathcal{X} \times \mathcal{Y}$ defined procedurally by first

---

**Algorithm 4** Agnostic PAC learner for $\mathcal{H}_i^j$ w.r.t. 0-1 loss

---

**Input:** Agnostic PAC learner $\mathcal{A}$ for $\mathcal{H}$ w.r.t $\ell$, unlabeled samples $S_U \sim \mathcal{D}_\mathcal{X}^n$, and labeled samples $S_L \sim \mathcal{D}^m$

1 For each $h \in \mathcal{H}_{|S_U}$, construct a dataset

$$S_U^h = \{(x_1, \tilde{y}_1), ..., (x_n, \tilde{y}_n)\} \text{ s.t. } \tilde{y}_i = \text{BinRel}(h(x_i), j)$$

2 Run $\mathcal{A}$ over all datasets to get $C(S_U) := \left\{ \mathcal{A}(S_U^h) \mid h \in \mathcal{H}_{|S_U} \right\}$

3 Define $C_i^j(S_U) = \{g_i^j | g \in C(S_U)\}$

4 Return $\hat{g}_i^j \in C_i^j(S_U)$ with the lowest empirical error over $S_L$ w.r.t. 0-1 loss.

---

sampling $x \sim \mathcal{D}_\mathcal{X}$ and then outputting the labeled sample $(x, \text{BinRel}(h^\star(x), j))$. Note that $\mathcal{D}^\star$ is a realizable distribution (realized by $h^\star$) w.r.t $\ell_{\text{sum}}^{@p}$ and therefore also $\ell$. Let $m_\mathcal{A}(\epsilon, \delta, K)$ be the sample complexity of $\mathcal{A}$. Since $\mathcal{A}$ is an agnostic PAC learner for $\mathcal{H}$ w.r.t $\ell$, we have that for sample size $n \geq m_\mathcal{A}(\frac{a\epsilon}{2}, \delta/2, K)$, with probability at least $1 - \frac{\delta}{2}$,

$$\mathbb{E}_{\mathcal{D}^\star} [\ell(g(x), y)] \leq \inf_{h \in \mathcal{H}} \mathbb{E}_{\mathcal{D}^\star} [\ell(h(x), y)] + \frac{a\epsilon}{2} = \frac{a\epsilon}{2}.$$

Furthermore, by definition of $\mathcal{D}^\star$, $\mathbb{E}_{\mathcal{D}^\star} [\ell(g(x), y)] = \mathbb{E}_{x \sim \mathcal{D}_\mathcal{X}} [\ell(g(x), \text{BinRel}(h^\star(x), j))]$. Therefore, $\mathbb{E}_{x \sim \mathcal{D}_\mathcal{X}} [\ell(g(x), \text{BinRel}(h^\star(x), j))] \leq \frac{a\epsilon}{2}$. Next, using Lemma E.3, we have pointwise that

$$
\begin{aligned}
\mathbb{1}\{g_i^j(x) \neq h_i^{\star,j}(x)\} &\leq \mathbb{1}\{\ell_{\text{sum}}^{@p}(g(x), \text{BinRel}(h^\star(x), j)) > 0\} \\
&= \mathbb{1}\{\ell(g(x), \text{BinRel}(h^\star(x), j)) > 0\} \\
&\leq \frac{1}{a} \ell(g(x), \text{BinRel}(h^\star(x), j)).
\end{aligned}
$$

Taking expectations on both sides gives,

$$\mathbb{E}_\mathcal{D} \left[ \mathbb{1}\{g_i^j(x) \neq h_i^{\star,j}(x)\} \right] \leq \frac{1}{a} \mathbb{E}_\mathcal{D} [\ell(g(x), \text{BinRel}(h^\star(x), j))] \leq \frac{\epsilon}{2},$$

where in the last inequality we use the fact that $\mathbb{E}_{x \sim \mathcal{D}_\mathcal{X}} [\ell(g(x), \text{BinRel}(h^\star(x), j))] \leq \frac{a\epsilon}{2}$. Finally, using the triangle inequality, we have that

$$
\begin{aligned}
\mathbb{E}_\mathcal{D} \left[ \mathbb{1}\{g_i^j(x) \neq y\} \right] &\leq \mathbb{E}_\mathcal{D} \left[ \mathbb{1}\{h_i^{\star,j}(x) \neq y\} \right] + \mathbb{E}_\mathcal{D} \left[ \mathbb{1}\{g_i^j(x) \neq h_i^{\star,j}(x)\} \right] \\
&\leq \mathbb{E}_\mathcal{D} \left[ \mathbb{1}\{h_i^{\star,j}(x) \neq y\} \right] + \frac{\epsilon}{2} \\
&= \arg\min_{h_i^j \in \mathcal{H}_i^j} \mathbb{E}_\mathcal{D} \left[ \mathbb{1}\{h_i^j(x) \neq y\} \right] + \frac{\epsilon}{2}.
\end{aligned}
$$

Since $g_i^j \in C_i^j(S_U)$, we have shown that $C_i^j(S_U)$ contains a hypothesis that generalizes well w.r.t $\mathcal{D}$. Now we want to show that the predictor $\hat{g}_i^j$ returned in step 4 also generalizes well. Crucially, observe that $C_i^j(S_U)$ is a finite hypothesis class with cardinality at most $K^{jn}$. Therefore, by standard Chernoff and union bounds, with probability at least $1 - \delta/2$, the empirical risk of every hypothesis in $C_i^j(S_U)$ on a sample of size $\geq \frac{8}{\epsilon^2} \log \frac{4|C_i^j(S_U)|}{\delta}$ is at most $\epsilon/4$ away from its true error. So, if $m = |S_L| \geq \frac{8}{\epsilon^2} \log \frac{4|C_i^j(S_U)|}{\delta}$, then with probability at least $1 - \delta/2$, we have

$$\frac{1}{|S_L|} \sum_{(x,y) \in S_L} \mathbb{1}\{g_i^j(x) \neq y\} \leq \mathbb{E}_\mathcal{D} \left[ \mathbb{1}\{g_i^j(x) \neq y\} \right] + \frac{\epsilon}{4} \leq \frac{3\epsilon}{4}.$$

Since $\hat{g}_i^j$ is the ERM on $S_L$ over $C_i^j(S_U)$, its empirical risk can be at most $\frac{3\epsilon}{4}$. Given that the population risk of $\hat{g}_i^j$ can be at most $\epsilon/4$ away from its empirical risk, we have that

$$\mathbb{E}_{\mathcal{D}}[\mathbb{1}\{\hat{g}_i^j(x) \neq y\}] \leq \underset{h_i^j \in \mathcal{H}_i^j}{\arg\min} \mathbb{E}_{\mathcal{D}}\left[\mathbb{1}\{h_i^j(x) \neq y\}\right] + \epsilon.$$

Applying union bounds, the entire process succeeds with probability $1 - \delta$. We can compute the upper bound on the sample complexity of Algorithm 4, denoted $n(\epsilon, \delta, K)$, as

$$n(\epsilon, \delta, K) \leq m_{\mathcal{A}}(\frac{a\epsilon}{2}, \delta/2, K) + O\left(\frac{1}{\epsilon^2} \log \frac{|C(S_U)|}{\delta}\right)$$

$$\leq m_{\mathcal{A}}(\frac{a\epsilon}{2}, \delta/2, K) + O\left(\frac{Km_{\mathcal{A}}(\frac{a\epsilon}{2}, \delta/2, K) + \log\frac{1}{\delta}}{\epsilon^2}\right),$$

where we use $|C(S_U)| \leq 2^{Km_{\mathcal{A}}(\frac{a\epsilon}{2}, \delta/2, K)}$. This shows that Algorithm 4 is an agnostic PAC learner for $\mathcal{H}_i^j$ w.r.t 0-1 loss. Since our choice of loss $\ell \in \mathcal{L}(\ell_{\text{sum}}^{@p})$ and indices $(i, j)$ were arbitrary, agnostic PAC learnability of $\mathcal{H}$ w.r.t $\ell$ implies agnostic PAC learnability of $\mathcal{H}_i^j$ w.r.t the 0-1 loss for all $(i, j) \in [K] \times [p]$. $\qquad\square$

## C.3    Characterizing Batch Learnability of $\mathcal{L}(\ell_{\text{prec}}^{@p})$

In this section, we prove Theorem 4.2 which characterizes the agnostic PAC learnability of an arbitrary hypothesis class $\mathcal{H} \subseteq \mathcal{S}_K^{\mathcal{X}}$ w.r.t losses in $\mathcal{L}(\ell_{\text{prec}}^{@p})$. Our proof will again be in three parts. First, we will show that if for all $i \in [K]$, $\mathcal{H}_i^p$ is agnostic PAC learnable w.r.t the 0-1 loss, then ERM is an agnostic PAC learnable w.r.t $\ell_{\text{prec}}^{@p}$. Next, we show that if $\mathcal{H}$ is agnostic PAC learnable w.r.t $\ell_{\text{prec}}^{@p}$, then $\mathcal{H}$ is agnostic PAC learnable w.r.t any loss $\ell \in \mathcal{L}(\ell_{\text{prec}}^{@p})$. Finally, we prove the necessity direction - if $\mathcal{H}$ is agnostic PAC learnable w.r.t an arbitrary $\ell \in \mathcal{L}(\ell_{\text{prec}}^{@p})$, then for all $i \in [K]$, $\mathcal{H}_i^p$ is agnostic PAC learnable w.r.t the 0-1 loss.

We begin with Lemma C.2 which asserts that if for all $i \in [K]$, $\mathcal{H}_i^p$ is agnostic PAC learnable, then ERM is an agnostic PAC learner for $\mathcal{H}$ w.r.t $\ell_{\text{prec}}^{@p}$.

**Lemma C.2.** *If for all $i \in [K]$, $\mathcal{H}_i^p$ is agnostic PAC learnable w.r.t the 0-1 loss, then ERM is an agnostic PAC learner for $\mathcal{H} \subseteq \mathcal{S}_K^{\mathcal{X}}$ w.r.t $\ell_{prec}^{@p}$*

The proof of Lemma C.2 is similar to the proof of Lemma 4.3 and involves bounding the empirical Rademacher complexity of the loss class $\ell_{\text{prec}}^{@p} \circ \mathcal{H}$. This will imply that $\ell_{\text{prec}}^{@p}$ enjoys the uniform convergence property, and therefore ERM is an agnostic PAC learner for $\mathcal{H}$ w.r.t $\ell_{\text{prec}}^{@p}$. The key insight is that we can write $\ell_{\text{prec}}^{@p}(h(x), y) = Z_y^p - \sum_{i=1}^K \mathbb{1}\{h_i(x) \leq p\}y^i = Z_y^p - \sum_{i=1}^K h_i^p(x)y^i$. Since $Z_y^p$ does not depend on $h(x)$ and $y^i \leq B$, we can upperbound the empirical Rademacher complexity in terms of the empirical Rademacher complexities of $\mathcal{H}_i^p$ using Talagrand's contraction.

*Proof.* Let $\mathcal{H} \subseteq \mathcal{S}_K^{\mathcal{X}}$ be an arbitrary ranking hypothesis class. Similar to the proof of Lemma 4.3, it suffices to show that the empirical Rademacher complexity of the loss class $\ell_{\text{prec}}^{@p} \circ \mathcal{H}$ vanishes. By Proposition C.1, this will imply that $\ell_{\text{prec}}^{@p}$ enjoys the uniform convergence property, and therefore

ERM is an agnostic PAC learner for $\mathcal{H}$ w.r.t $\ell_{\text{prec}}^{@p}$. By definition, we have that

$$
\begin{aligned}
\hat{\mathfrak{R}}_n(\ell_{\text{prec}}^{@p} \circ \mathcal{H}) &= \mathbb{E}_{\sigma \sim \{\pm 1\}^n}\left[\sup_{h \in \mathcal{H}} \frac{1}{n} \sum_{i=1}^{n} \sigma_i \ell_{\text{prec}}^{@p}(h(x_i), y_i))\right] \\
&= \mathbb{E}_{\sigma \sim \{\pm 1\}^n}\left[\sup_{h \in \mathcal{H}} \frac{1}{n} \sum_{i=1}^{n} \left(\sigma_i Z_{y_i}^p - \sum_{m=1}^{K} \sigma_i \mathbb{1}\{h_m(x_i) \le p\} y_i^m\right)\right] \\
&= \mathbb{E}_{\sigma \sim \{\pm 1\}^n}\left[\sup_{h \in \mathcal{H}} \frac{1}{n} \sum_{i=1}^{n} \sum_{m=1}^{K} \sigma_i h_m^p(x_i) y_i^m\right] \\
&\le \sum_{m=1}^{K} \mathbb{E}_{\sigma \sim \{\pm 1\}^n}\left[\sup_{h \in \mathcal{H}} \frac{1}{n} \sum_{i=1}^{n} \sigma_i h_m^p(x_i) y_i^m\right] \\
&\le B \sum_{m=1}^{K} \mathbb{E}_{\sigma \sim \{\pm 1\}^n}\left[\sup_{h \in \mathcal{H}} \frac{1}{n} \sum_{i=1}^{n} \sigma_i h_m^p(x_i)\right] \\
&= B \sum_{m=1}^{K} \hat{\mathfrak{R}}_n(\mathcal{H}_m^p),
\end{aligned}
$$

where the second inequality follows from Talagrand's Contraction Lemma and the fact that $y_i^m \le B$ for all $i, m$. Since for all $m \in [K]$, $\mathcal{H}_m^p$ is agnostic PAC learnable w.r.t 0-1 loss, by Theorem 6.7 in Shalev-Shwartz and Ben-David [2014], $\lim_{n \to \infty} \hat{\mathfrak{R}}_n(\mathcal{H}_m^p) = 0$. Since $K$ and $B$ are finite,

$$
\lim_{n \to \infty} \hat{\mathfrak{R}}_n(\ell_{\text{prec}}^{@p} \circ \mathcal{H}) = \lim_{n \to \infty} B \sum_{m=1}^{K} \hat{\mathfrak{R}}_n(\mathcal{H}_m^p) = 0
$$

.

By Proposition C.1, this implies that $\ell_{\text{prec}}^{@p}$ enjoys the uniform convergence property, and therefore ERM using $\ell_{\text{prec}}^{@p}$ is an agnostic PAC learner for $\mathcal{H}$. $\qquad\square$

Next, Lemma C.3 extends the learnability of $\ell_{\text{prec}}^{@p}$ to the learnability of any loss $\ell \in \mathcal{L}(\ell_{\text{prec}}^{@p})$. In particular, Lemma C.3 asserts that if $\mathcal{H}$ is agnostic PAC learnable w.r.t $\ell_{\text{prec}}^{@p}$ then $\mathcal{H}$ is also agnostic PAC learnable w.r.t any $\ell \in \mathcal{L}(\ell_{\text{prec}}^{@p})$.

**Lemma C.3.** *If a hypothesis class $\mathcal{H} \subseteq \mathcal{S}_K^{\mathcal{X}}$ is agnostic PAC learnable w.r.t $\ell_{\text{prec}}^{@p}$, then $\mathcal{H}$ is agnostic PAC learnable w.r.t any $\ell \in \mathcal{L}(\ell_{\text{prec}}^{@p})$.*

The proof of Lemma C.3 follows the same the exact same strategy used in proving Lemma 4.4. More specifically, given an agnostic PAC learner $\mathcal{A}$ for $\mathcal{H}$ w.r.t. $\ell_{\text{prec}}^{@p}$, we first create a *realizable* PAC learner for $\mathcal{H}$ w.r.t $\ell \in \mathcal{L}(\ell_{\text{prec}}^{@p})$. Then, we use a similar realizable-to-agnostic conversion technique as in the proof of Lemma 4.4 to convert the realizable PAC learner into an agnostic PAC learner for $\mathcal{H}$ w.r.t $\ell$.

*Proof.* Fix $\ell \in \mathcal{L}(\ell_{\text{prec}}^{@p})$. Let $a = \min_{\pi, y}\{\ell(\pi, y) \mid \ell(\pi, y) \ne 0\}$ and $b = \max_{\pi, y} \ell(\pi, y)$. We need to show that if $\mathcal{H}$ is agnostic PAC learnable w.r.t $\ell_{\text{prec}}^{@p}$, then $\mathcal{H}$ is agnostic PAC learnable w.r.t $\ell$. We will do so in two steps. First, we will show that if $\mathcal{A}$ is an agnostic PAC learner for $\mathcal{H}$ w.r.t. $\ell_{\text{prec}}^{@p}$, then $\mathcal{A}$ is also a *realizable* PAC learner for $\mathcal{H}$ w.r.t $\ell$. Next, we will show how to convert the realizable PAC learner w.r.t $\ell$ into an agnostic PAC learner w.r.t $\ell$ in a black-box fashion. The composition of these two pieces yields an agnostic PAC learner for $\mathcal{H}$ w.r.t $\ell$.

If $\mathcal{H}$ is agnostic PAC learnable w.r.t $\ell_{\text{prec}}^{@p}$, then there exists a learning algorithm $\mathcal{A}$ with sample complexity $m(\epsilon, \delta, K)$ s.t. for any distribution $\mathcal{D}$ over $\mathcal{X} \times \mathcal{Y}$, with probability $1 - \delta$ over a sample $S \sim \mathcal{D}^n$ of size $n \ge m(\epsilon, \delta, K)$, the output $g = \mathcal{A}(S)$ achieves

$$\mathbb{E}_{\mathcal{D}}\left[\ell_{\mathrm{prec}}^{@p}(g(x),y))\right] \leq \inf_{h\in\mathcal{H}}\mathbb{E}_{\mathcal{D}}\left[\ell_{\mathrm{prec}}^{@p}(h(x),y))\right] + \epsilon.$$

If $\mathcal{D}$ is realizable w.r.t $\ell$, then we are guaranteed that there exists a hypothesis $h^\star \in \mathcal{H}$ s.t. $\mathbb{E}_{\mathcal{D}}\left[\ell(h^\star(x),y)\right] = 0$. Since $\ell \in \mathcal{L}(\ell_{\mathrm{prec}}^{@p})$, this also means that $\mathbb{E}_{\mathcal{D}}\left[\ell_{\mathrm{prec}}^{@p}(h^\star(x),y)\right] = 0$. Furthermore, since $\ell \in \mathcal{L}(\ell_{\mathrm{prec}}^{@p})$, $\ell \leq b\ell_{\mathrm{prec}}^{@p}$. Together, this means we have $\mathbb{E}_{\mathcal{D}}\left[\ell(g(x),y)\right] \leq b\epsilon$ showing have that $\mathcal{A}$ is also a realizable PAC learner for $\mathcal{H}$ w.r.t $\ell$ with sample complexity $m(\frac{\epsilon}{b},\delta,K)$. This completes the first part of the proof.

Now, we show how to convert the realizable PAC learner $\mathcal{A}$ for $\ell$ into an agnostic PAC learner for $\ell$ in a black-box fashion. For this step, we will use a similar algorithm as in the proof of Lemma 4.4. That is, we will show that Algorithm 5 below is an agnostic PAC learner for $\mathcal{H}$ w.r.t $\ell$.

---

**Algorithm 5** Agnostic PAC learner for $\mathcal{H}$ w.r.t. $\ell$

---

**Input:** Realizable PAC learner $\mathcal{A}$ for $\mathcal{H}$ w.r.t $\ell$, unlabeled samples $S_U \sim \mathcal{D}_{\mathcal{X}}^n$, and labeled samples $S_L \sim \mathcal{D}^m$

1 For each $h \in \mathcal{H}_{|S_U}$, construct a dataset
$$S_U^h = \{(x_1,\tilde{y}_1),...,(x_n,\tilde{y}_n)\} \text{ s.t. } \tilde{y}_i = \mathrm{BinRel}(h(x_i),p)$$

2 Run $\mathcal{A}$ over all datasets to get $C(S_U) := \left\{\mathcal{A}(S_U^h) \mid h \in \mathcal{H}_{|S_U}\right\}$

3 Return $\hat{g} \in C(S_U)$ with the lowest empirical error over $S_L$ w.r.t. $\ell$.

---

Let $\mathcal{D}$ be any (not necessarily realizable) distribution over $\mathcal{X} \times \mathcal{Y}$. Let $h^\star = \arg\min_{h\in\mathcal{H}} \mathbb{E}_{\mathcal{D}}\left[\ell(h(x),y))\right]$ denote the optimal predictor in $\mathcal{H}$ w.r.t $\mathcal{D}$. Consider the sample $S_U^{h^\star}$ and let $g = \mathcal{A}(S_U^{h^\star})$. We can think of $g$ as the output of $\mathcal{A}$ run over an i.i.d sample $S$ drawn from $\mathcal{D}^\star$, a joint distribution over $\mathcal{X} \times \mathcal{Y}$ defined procedurally by first sampling $x \sim \mathcal{D}_{\mathcal{X}}$, and then outputting the labeled sample $(x, \mathrm{BinRel}(h^\star(x),p))$. Note that $\mathcal{D}^\star$ is indeed a realizable distribution (realized by $h^\star$) w.r.t both $\ell$ and $\ell_{\mathrm{prec}}^{@p}$. Recall that $m_{\mathcal{A}}(\frac{\epsilon}{b},\delta,K)$ is the sample complexity of $\mathcal{A}$. Since $\mathcal{A}$ is a realizable learner for $\mathcal{H}$ w.r.t $\ell$, we have that for $n \geq m_{\mathcal{A}}(\frac{a\epsilon}{2b^2},\delta/2,K)$, with probability at least $1 - \frac{\delta}{2}$,

$$\mathbb{E}_{\mathcal{D}^\star}\left[\ell(g(x),y)\right] \leq \frac{a\epsilon}{2b}.$$

By definition of $\mathcal{D}^\star$, it further follows that $\mathbb{E}_{\mathcal{D}^\star}\left[\ell(g(x),y)\right] = \mathbb{E}_{x\sim\mathcal{D}_{\mathcal{X}}}\left[\ell(g(x),\mathrm{BinRel}(h^\star(x),p))\right]$. Therefore,

$$\mathbb{E}_{x\sim\mathcal{D}_{\mathcal{X}}}\left[\ell(g(x),\mathrm{BinRel}(h^\star(x),p))\right] \leq \frac{a\epsilon}{2b}.$$

Next, by Lemma E.2, we have pointwise that:

$$\ell(g(x),y) \leq \ell(h^\star(x),y) + \frac{b}{a}\ell(g(x),\mathrm{BinRel}(h^\star(x),p)).$$

Taking expectations on both sides of the inequality gives:

$$\mathbb{E}_{\mathcal{D}}\left[\ell(g(x),y)\right] \leq \mathbb{E}_{\mathcal{D}}\left[\ell(h^\star(x),y)\right] + \mathbb{E}_{\mathcal{D}}\left[\frac{b}{a}\ell(g(x),\mathrm{BinRel}(h^\star(x),p))\right]$$
$$= \mathbb{E}_{\mathcal{D}}\left[\ell(h^\star(x),y)\right] + \frac{b}{a}\mathbb{E}_{x\sim\mathcal{D}_{\mathcal{X}}}\left[\ell(g(x),\mathrm{BinRel}(h^\star(x),p))\right]$$
$$\leq \mathbb{E}_{\mathcal{D}}\left[\ell(h^\star(x),y)\right] + \frac{\epsilon}{2}.$$

Therefore, we have shown that $C(S_U)$ contains a hypothesis $g$ that generalizes well with respect to $\mathcal{D}$. The remaining proof follows exactly as in the proof of Lemma 4.4. We include them here for the sake of completeness.

Now we want to show that the predictor $\hat{g}$ returned in step 4 also has good generalization. Crucially, observe that $C(S_U)$ is a finite hypothesis class with cardinality at most $K^{pn}$. Therefore, by standard Chernoff and union bounds, with probability at least $1 - \delta/2$, the empirical risk of every hypothesis in $C(S_U)$ on a sample of size $\geq \frac{8}{\epsilon^2} \log \frac{4|C(S_U)|}{\delta}$ is at most $\epsilon/4$ away from its true error. So, if $m = |S_L| \geq \frac{8}{\epsilon^2} \log \frac{4|C(S_U)|}{\delta}$, then with probability at least $1 - \delta/2$, we have

$$\frac{1}{|S_L|} \sum_{(x,y) \in S_L} \ell(g(x), y) \leq \mathbb{E}_{\mathcal{D}}\left[\ell(g(x), y)\right] + \frac{\epsilon}{4} \leq \mathbb{E}_{\mathcal{D}}\left[\ell(h^\star(x), y)\right] + \frac{3\epsilon}{4}.$$

Since $\hat{g}$ is the ERM on $S_L$ over $C(S)$, its empirical risk can be at most $\mathbb{E}_{\mathcal{D}}\left[\ell(h^\star(x), y)\right] + \frac{3\epsilon}{4}$. Given that the population risk of $\hat{g}$ can be at most $\epsilon/4$ away from its empirical risk, we have that

$$\mathbb{E}_{\mathcal{D}}[\ell(\hat{g}(x), y)] \leq \mathbb{E}_{\mathcal{D}}\left[\ell(h^\star(x), y)\right] + \epsilon.$$

Applying union bounds, the entire process succeeds with probability $1 - \delta$. We can upper bound the sample complexity of Algorithm 1, denoted $n(\epsilon, \delta, K)$, as

$$n(\epsilon, \delta, K) \leq m_{\mathcal{A}}(\frac{a\epsilon}{2b^2}, \delta/2, K) + O\left(\frac{1}{\epsilon^2} \log \frac{|C(S_U)|}{\delta}\right)$$

$$\leq m_{\mathcal{A}}(\frac{a\epsilon}{2b^2}, \delta/2, K) + O\left(\frac{p\, m_{\mathcal{A}}(\frac{a\epsilon}{2b^2}, \delta/2, K) \log(K) + \log \frac{1}{\delta}}{\epsilon^2}\right),$$

where we use $|C(S_U)| \leq K^{pm_{\mathcal{A}}(\frac{a\epsilon}{2b^2}, \delta/2, K)}$. This shows that Algorithm 1, given as input an realizable PAC learner for $\mathcal{H}$ w.r.t $\ell$, is an agnostic PAC learner for $\mathcal{H}$ w.r.t $\ell$. Using the realizable learner we constructed before this step as the input completes this proof as we have constructively converted an agnostic PAC learner for $\ell_{\text{prec}}^{@p}$ into an agnostic PAC learner for $\ell$. $\qquad\square$

Lemma C.2 and C.3 together complete the proof of sufficiency in Theorem 4.2. Finally, Lemma C.4 below shows that the agnostic PAC learnability of $\mathcal{H}_i^p$ for all $i \in [K]$ is necessary for the agnostic PAC learnability of $\mathcal{H}$ w.r.t any $\ell \in \mathcal{L}(\ell_{\text{prec}}^{@p})$. Like before, the proof of Lemma C.4 is constructive and follows exactly the same strategy as Lemma 4.5. That is, given as input a learner for $\ell$, we will convert it into an agnostic learner for $\mathcal{H}_i^p$. In fact, the conversion is exactly the same as in the proof of Lemma 4.5 and just requires running Algorithm 4 with an input learner for $\ell \in \mathcal{L}(\ell_{\text{prec}}^{@p})$ and setting $j = p$.

**Lemma C.4.** *If a function class $\mathcal{H} \subseteq \mathcal{S}_K^{\mathcal{X}}$ is agnostic PAC learnable w.r.t $\ell \in \mathcal{L}(\ell_{prec}^{@p})$, then $\mathcal{H}_i^p$ is agnostic PAC learnable w.r.t the 0-1 loss for all $i \in [K]$.*

*Proof.* Fix $\ell \in \mathcal{L}(\ell_{\text{prec}}^{@p})$ and $i \in [K]$. Let $a = \min_{\pi, y}\{\ell(\pi, y) \mid \ell(\pi, y) \neq 0\}$. Let $\mathcal{H}$ be an arbitrary ranking hypothesis class and $\mathcal{A}$ be an agnostic PAC learner for $\mathcal{H}$ w.r.t $\ell$. Our goal will to be to use $\mathcal{A}$ to construct an agnostic PAC learner for $\mathcal{H}_i^p$.

Let $\mathcal{D}$ be any distribution over $\mathcal{X} \times \{0, 1\}$, $h_i^{\star,p} = \arg\min_{h \in \mathcal{H}_i^p} \mathbb{E}_{\mathcal{D}}\left[\mathbb{1}\{h(x) \neq y\}\right]$ the optimal hypothesis, and $h^\star \in \mathcal{H}$ be any valid completion of $h_i^{\star,p}$. We will now show that Algorithm 4 from the proof of Lemma 4.5 is an agnostic PAC learner for $\mathcal{H}_i^p$ if we set $j = p$ and give it as input an agnostic PAC learner $\mathcal{A}$ for $\mathcal{H}$ w.r.t. $\ell \in \mathcal{L}(\ell_{\text{prec}}^{@p})$.

Consider the sample $S_U^{h^\star}$ and let $g = \mathcal{A}(S_U^{h^\star})$. We can think of $g$ as the output of $\mathcal{A}$ run over an i.i.d sample $S$ drawn from $\mathcal{D}^\star$, a joint distribution over $\mathcal{X} \times \mathcal{Y}$ defined procedurally by first sampling $x \sim \mathcal{D}_{\mathcal{X}}$ and then outputting the labeled sample $(x, \text{BinRel}(h^\star(x), p))$. Note that $\mathcal{D}^\star$ is a realizable distribution (realized by $h^\star$) w.r.t $\ell_{\text{prec}}^{@p}$ and therefore also $\ell$. Let $m_{\mathcal{A}}(\epsilon, \delta, K)$ be the sample complexity of $\mathcal{A}$.

Since $\mathcal{A}$ is an agnostic PAC learner for $\mathcal{H}$ w.r.t $\ell$, we have that for sample size $n \geq m_{\mathcal{A}}(\frac{a\epsilon}{2}, \delta/2, K)$, with probability at least $1 - \frac{\delta}{2}$,

$$\mathbb{E}_{\mathcal{D}^\star}\left[\ell(g(x),y)\right] \leq \inf_{h\in\mathcal{H}} \mathbb{E}_{\mathcal{D}^\star}\left[\ell(h(x),y)\right] + \frac{a\epsilon}{2} = \frac{a\epsilon}{2}.$$

Furthermore, by definition of $\mathcal{D}^\star$, $\mathbb{E}_{\mathcal{D}^\star}\left[\ell(g(x),y)\right] = \mathbb{E}_{x\sim\mathcal{D}_\mathcal{X}}\left[\ell(g(x),\mathrm{BinRel}(h^\star(x),p))\right]$. Therefore, $\mathbb{E}_{x\sim\mathcal{D}_\mathcal{X}}\left[\ell(g(x),\mathrm{BinRel}(h^\star(x),p))\right] \leq \frac{a\epsilon}{2}$. Next, using Lemma E.4, we have pointwise that

$$
\begin{aligned}
\mathbb{1}\{g_i^p(x) \neq h_i^{\star,p}(x)\} &\leq \mathbb{1}\{\ell_{\mathrm{prec}}^{@p}(g(x),\mathrm{BinRel}(h^\star(x),p)) > 0\} \\
&= \mathbb{1}\{\ell(g(x),\mathrm{BinRel}(h^\star(x),p)) > 0\} \\
&\leq \frac{1}{a}\,\ell(g(x),\mathrm{BinRel}(h^\star(x),p)).
\end{aligned}
$$

Taking expectations on both sides gives,

$$\mathbb{E}_\mathcal{D}\left[\mathbb{1}\{g_i^p(x) \neq h_i^{\star,p}(x)\}\right] \leq \frac{1}{a}\mathbb{E}_\mathcal{D}\left[\ell(g(x),\mathrm{BinRel}(h^\star(x),p))\right] \leq \frac{\epsilon}{2},$$

where in the last inequality we use the fact that $\mathbb{E}_{x\sim\mathcal{D}_\mathcal{X}}\left[\ell(g(x),\mathrm{BinRel}(h^\star(x),p))\right] \leq \frac{a\epsilon}{2}$. Finally, using the triangle inequality, we have that

$$
\begin{aligned}
\mathbb{E}_\mathcal{D}\left[\mathbb{1}\{g_i^p(x) \neq y\}\right] &\leq \mathbb{E}_\mathcal{D}\left[\mathbb{1}\{h_i^{\star,p}(x) \neq y\}\right] + \mathbb{E}_\mathcal{D}\left[\mathbb{1}\{g_i^p(x) \neq h_i^{\star,p}(x)\}\right] \\
&\leq \mathbb{E}_\mathcal{D}\left[\mathbb{1}\{h_i^{\star,p}(x) \neq y\}\right] + \frac{\epsilon}{2} \\
&= \operatorname*{arg\,min}_{h_i^p \in \mathcal{H}_i^p} \mathbb{E}_\mathcal{D}\left[\mathbb{1}\{h_i^p(x) \neq y\}\right] + \frac{\epsilon}{2}.
\end{aligned}
$$

Since $g_i^p \in C_i^p(S_U)$, we have shown that $C_i^p(S_U)$ contains a hypothesis that generalizes well w.r.t $\mathcal{D}$. Now we want to show that the predictor $\hat{g}_i^p$ returned in step 4 also generalizes well. Crucially, observe that $C_i^p(S_U)$ is a finite hypothesis class with cardinality at most $K^{pn}$. Therefore, by standard Chernoff and union bounds, with probability at least $1 - \delta/2$, the empirical risk of every hypothesis in $C_i^p(S_U)$ on a sample of size $\geq \frac{8}{\epsilon^2}\log\frac{4|C_i^j(S_U)|}{\delta}$ is at most $\epsilon/4$ away from its true error. So, if $m = |S_L| \geq \frac{8}{\epsilon^2}\log\frac{4|C_i^j(S_U)|}{\delta}$, then with probability at least $1 - \delta/2$, we have

$$\frac{1}{|S_L|}\sum_{(x,y)\in S_L}\mathbb{1}\{g_i^p(x) \neq y\} \leq \mathbb{E}_\mathcal{D}\left[\mathbb{1}\{g_i^p(x) \neq y\}\right] + \frac{\epsilon}{4} \leq \frac{3\epsilon}{4}.$$

Since $\hat{g}_i^p$ is the ERM on $S_L$ over $C_i^p(S_U)$, its empirical risk can be at most $\frac{3\epsilon}{4}$. Given that the population risk of $\hat{g}_i^p$ can be at most $\epsilon/4$ away from its empirical risk, we have that

$$\mathbb{E}_\mathcal{D}[\mathbb{1}\{\hat{g}_i^p(x) \neq y\}] \leq \operatorname*{arg\,min}_{h_i^p \in \mathcal{H}_i^p} \mathbb{E}_\mathcal{D}\left[\mathbb{1}\{h_i^p(x) \neq y\}\right] + \epsilon.$$

Applying union bounds, the entire process succeeds with probability $1 - \delta$. We can compute the upper bound on the sample complexity of Algorithm 4, denoted $n(\epsilon,\delta,K)$, as

$$
\begin{aligned}
n(\epsilon,\delta,K) &\leq m_\mathcal{A}(\frac{a\epsilon}{2},\delta/2,K) + O\left(\frac{1}{\epsilon^2}\log\frac{|C(S_U)|}{\delta}\right) \\
&\leq m_\mathcal{A}(\frac{a\epsilon}{2},\delta/2,K) + O\left(\frac{p\,m_\mathcal{A}(\frac{a\epsilon}{2},\delta/2,K)\log(K) + \log\frac{1}{\delta}}{\epsilon^2}\right),
\end{aligned}
$$

where we use $|C(S_U)| \leq K^{pm_\mathcal{A}(\frac{a\epsilon}{2},\delta/2,K)}$. This shows that Algorithm 4 is an agnostic PAC learner for $\mathcal{H}_i^p$ w.r.t 0-1 loss. Since our choice of loss $\ell \in \mathcal{L}(\ell_{\mathrm{prec}}^{@p})$ and index $i$ were arbitrary, agnostic PAC learnability of $\mathcal{H}$ w.r.t $\ell$ implies agnostic PAC learnability of $\mathcal{H}_i^p$ w.r.t the 0-1 loss for all $i \in [K]$. $\qquad\square$

Combining Lemma C.2, C.3 and C.4 gives Theorem 4.2.

# D  Proofs for Online Multilabel Ranking

## D.1  Proof of necessity in Theorem 5.1

*Proof.* Fix $\ell \in \mathcal{L}(\ell_{\text{sum}}^{@p})$ and $(i,j) \in [K] \times [p]$. Given an online learner $\mathcal{A}$ for $\mathcal{H}$ w.r.t $\ell$, our goal is to construct an agnostic online learner $\mathcal{A}_i^j$ for $\mathcal{H}_i^j$. To that end, let $(x_1, y_1), ..., (x_T, y_T) \in (\mathcal{X} \times \{0,1\})^T$ denote a stream of labeled instances. Define $h_i^{\star,j} = \arg\min_{h_i^j \in \mathcal{H}_i^j} \sum_{t=1}^T \mathbb{1}\{h_i^j(x_t) \neq y_t\}$ to be the optimal function in $\mathcal{H}_i^j$ and $h^\star$ be an arbitrary completion of $h_i^{\star,j}$. As in the sufficiency proof, our construction of the online learner for $\mathcal{H}_i^j$ will run REWA over a set of experts we construct below.

For any bitstring $b \in \{0,1\}^T$, let $\phi : \{t \in [T] : b_t = 1\} \to \mathcal{S}_K$ denote a function mapping time points where $b_t = 1$ to permutations. Let $\Phi_b = \mathcal{S}_K^{\{t \in [T]: b_t = 1\}}$ denote all such functions $\phi$. For every $h \in \mathcal{H}$, there exists a $\phi_b^h \in \Phi_b$ such that for all $t \in \{t : b_t = 1\}$, $\phi_b^h(t) = h(x_t)$. Let $|b| = |\{t \in [T] : b_t = 1\}|$. For every $b \in \{0,1\}^T$ and $\phi \in \Phi_b$, define an Expert $E_{b,\phi}$. Expert $E_{b,\phi}$, formally presented in Algorithm 6, uses $\mathcal{A}$ to make predictions in each round. For every $b \in \{0,1\}^T$, let $\mathcal{E}_b = \bigcup_{\phi \in \Phi_b} \{E_{b,\phi}\}$ denote the set of all Experts parameterized by functions $\phi \in \Phi_b$. As before, we will actually define $\mathcal{E}_b = \{E_0\} \cup \bigcup_{\phi \in \Phi_b} \{E_{b,\phi}\}$, where $E_0$ is the expert that never updates $\mathcal{A}$ and only uses it to make predictions in each round. Note that $1 \leq |\mathcal{E}_b| \leq (K!)^{|b|} \leq K^{K|b|}$.

---

**Algorithm 6** Expert $(b, \phi)$

**Input:** Independent copy of online learner $\mathcal{A}$ for $\mathcal{H}$
1  **for** $t = 1, ..., T$ **do**
2  $\quad$ Receive example $x_t$
3  $\quad$ Predict $\mathbb{1}\{\hat{\pi}_i \leq j\}$ where $\hat{\pi} = \mathcal{A}(x_t)$
4  $\quad$ **if** $b_t = 1$ **then**
5  $\quad\quad$ Update $\mathcal{A}$ by passing $(x_t, \text{BinRel}(\phi(t), j))$
6  **end**

---

We are now ready to give the agnostic online learner for $\mathcal{H}_i^j$, henceforth denoted by $\mathcal{Q}$. Our online learner $\mathcal{Q}$ is very similar to Algorithm 3. First, it will sample a $B \in \{0,1\}^T$ s.t. $B_t \sim$ Bernoulli$(T^\beta/T)$. Then, it will construct a set of experts $\mathcal{E}_B$ using Algorithm 6. Finally, it will run REWA, denoted by $\mathcal{P}$, on the 0-1 loss over the stream $(x_1, y_1), ..., (x_T, y_T)$. As before, let $A$ and $P$ be the random variables denoting internal randomness of the algorithm $\mathcal{A}$ and $\mathcal{P}$. Using REWA guarantees and following exactly the same calculation as in the sufficiency proof, we arrive at

$$\mathbb{E}\left[\sum_{t=1}^T \mathbb{1}\{\mathcal{Q}(x_t) \neq y_t\}\right] \leq \mathbb{E}\left[\sum_{t=1}^T \mathbb{1}\{E_{B,\phi_B^{h^\star}}(x_t) \neq y_t\}\right] + \sqrt{2T^{1+\beta}K\ln K}.$$

The inequality above is the adaptation of Equation (1) for this proof. Recall that $h_i^{\star,j}$ is the optimal function in hindsight for the stream and $h^\star$ is a completion of $h_i^{\star,j}$. Since $\mathbb{1}\{E_{B,\phi_B^{h^\star}}(x_t) \neq y_t\} \leq \mathbb{1}\{h_i^{\star,j}(x_t) \neq y_t\} + \mathbb{1}\{E_{B,\phi_B^{h^\star}}(x_t) \neq h_i^{\star,j}(x_t)\}$, the inequality above reduces to

$$\mathbb{E}\left[\sum_{t=1}^T \mathbb{1}\{\mathcal{Q}(x_t) \neq y_t\}\right] \leq \sum_{t=1}^T \mathbb{1}\{h_i^{\star,j}(x_t) \neq y_t\} + \mathbb{E}\left[\sum_{t=1}^T \mathbb{1}\{E_{B,\phi_B^{h^\star}}(x_t) \neq h_i^{\star,j}(x_t)\}\right] + \sqrt{2T^{1+\beta}K\ln K}.$$

It now suffices to show that $\mathbb{E}\left[\sum_{t=1}^T \mathbb{1}\{E_{B,\phi_B^{h^\star}}(x_t) \neq h_i^{\star,j}(x_t)\}\right]$ is sub-linear function of $T$.

Given an online learner $\mathcal{A}$ for $\mathcal{H}$, an instance $x \in \mathcal{X}$, and an ordered finite sequence of labeled examples $L \in (\mathcal{X} \times \mathcal{Y})^*$, let $\mathcal{A}(x|L)$ be the random variable denoting the prediction of $\mathcal{A}$ on the instance $x$ after running and updating on $L$. For any $b \in \{0,1\}^T$, $h \in \mathcal{H}$, and $t \in [T]$, let $L_{b<t}^h = \{(x_i, \text{BinRel}(h(x_s), j)) : s < t \text{ and } b_s = 1\}$ denote the *subsequence* of the sequence of

labeled instances $\{(x_s, \mathrm{BinRel}(h(x_s), j))\}_{s=1}^{t-1}$ where $b_s = 1$. Thus, using Lemma E.3, we have

$$\mathbb{1}\{E_{B,\phi_B^{h^\star}}(x_t) \neq h_i^{\star,j}(x_t)\} \leq \mathbb{1}\{\ell_{\mathrm{sum}}^{@p}(\mathcal{A}(x_t \mid L_{B_{<t}}^{h^\star}), \mathrm{BinRel}(h^\star(x_t), j)) > 0\}$$
$$= \mathbb{1}\{\ell(\mathcal{A}(x_t \mid L_{B_{<t}}^{h^\star}), \mathrm{BinRel}(h^\star(x_t), j)) > 0\}$$
$$\leq \frac{1}{a}\ell(\mathcal{A}(x_t \mid L_{B_{<t}}^{h^\star}), \mathrm{BinRel}(h^\star(x_t), j), \mathrm{BinRel}(h^\star(x_t), j)),$$

where equality follows from the fact that $\ell \in \mathcal{L}(\ell_{\mathrm{sum}}^{@p})$. Here, $a$ is the lower bound whenever it is non-zero. Taking expectations of both sides and summing over $t \in [T]$ gives

$$\mathbb{E}\left[\sum_{t=1}^T \mathbb{1}\{E_{B,\phi_B^{h^\star}}(x_t) \neq h_i^{\star,j}(x_t)\}\right] \leq \frac{1}{a}\mathbb{E}\left[\sum_{t=1}^T \ell(\mathcal{A}(x_t \mid L_{B_{<t}}^{h^\star}), \mathrm{BinRel}(h^\star(x_t), j))\right].$$

To upperbound the right-hand side, we will again use the fact that the prediction $\mathcal{A}(x_t \mid L_{B_{<t}}^{h^\star})$ only depends on $(B_1, \ldots, B_{t-1})$, but is independent of $B_t$. The details of this calculation are omitted because they are identical to that of the sufficiency proof. Using independence of $\mathcal{A}(x_t \mid L_{B_{<t}}^{h^\star})$ and $B_t$, we obtain

$$\mathbb{E}\left[\sum_{t=1}^T \ell(\mathcal{A}(x_t \mid L_{B_{<t}}^{h^\star}), \mathrm{BinRel}(h^\star(x_t), j))\right] = \frac{T}{T^\beta}\mathbb{E}\left[\sum_{t:B_t=1} \ell(\mathcal{A}(x_t \mid L_{B_{<t}}^{h^\star}), \mathrm{BinRel}(h^\star(x_t), j))\right]$$
$$= \frac{T}{T^\beta}\mathbb{E}\left[\mathbb{E}\left[\sum_{t:B_t=1} \ell(\mathcal{A}(x_t \mid L_{B_{<t}}^{h^\star}), \mathrm{BinRel}(h^\star(x_t), j)) \Big| B\right]\right]$$
$$\leq \frac{T}{T^\beta}\mathbb{E}\left[R(|B|, K)\right],$$

where $R(|B|, K)$ is the regret of the algorithm $\mathcal{A}$, a sub-linear function of $|B|$. In the last step, we use the fact that $\mathcal{A}$ is a (realizable) online learner for $\mathcal{H}$ w.r.t. $\ell$ and the feedback that the algorithm received was $(x_t, \mathrm{BinRel}(h^\star(x_t), j))$ in the rounds whenever $B_t = 1$. Again, Lemma 5.17 from Ceccherini-Silberstein et al. [2017] guarantees an existence of a concave sublinear upperbound $\tilde{R}(|B|, K)$ of $R(|B|, K)$. Then, applying Jensen's inequality yields $\mathbb{E}[R(|B|, K)] \leq \mathbb{E}\left[\tilde{R}(|B|, K)\right] \leq \tilde{R}(T^\beta, K)$, a concave sub-linear function of $T^\beta$. Combining everything, we get

$$\mathbb{E}\left[\sum_{t=1}^T \mathbb{1}\{\mathcal{Q}(x_t) \neq y_t\}\right] \leq \sum_{t=1}^T \mathbb{1}\{h_i^{\star,j}(x_t) \neq y_t\} + \frac{T}{aT^\beta}\tilde{R}(T^\beta, K) + \sqrt{2T^{1+\beta}K\ln K}$$
$$= \arg\min_{h_i^j \in \mathcal{H}_i^j}\sum_{t=1}^T \mathbb{1}\{h_i^j(x_t) \neq y_t\} + \frac{T}{aT^\beta}\tilde{R}(T^\beta, K) + \sqrt{2T^{1+\beta}K\ln K}$$

For any choice of $\beta \in (0, 1)$, the regret above is a sub-linear function of $T$. Therefore, we have shown that $\mathcal{Q}$ is an agnostic learner for $\mathcal{H}_i^j$ w.r.t. 0-1 loss. $\qquad\square$

### D.2 Proof of Theorem 5.2

*Proof.* (of sufficiency in Theorem 5.2) Fix $\ell \in \mathcal{L}(\ell_{\mathrm{prec}}^{@p})$ and let $M = \max_{\pi,y}\ell(\pi, y)$. This proof is virtually identical to the proof of sufficiency in Theorem 4.1. However, we provide the full details here for completion. Our proof is also based on reduction. That is, given realizable learners $\mathcal{A}_i^p$ of $\mathcal{H}_i^p$'s for $i \in [K]$ w.r.t. 0-1 loss, we will construct an agnostic learner $\mathcal{Q}$ for $\mathcal{H}$ w.r.t. $\ell$. We will construct a set of experts $\mathcal{E}$ that uses $\mathcal{A}_i^p$ to make predictions and run the REWA algorithm using these experts.

Let $(x_1, y_1), \ldots, (x_T, y_T) \in (\mathcal{X} \times \mathcal{Y})^T$ denote the stream of points to be observed by the online learner. As before, we will assume an oblivious adversary. Define $h^\star = \arg\min_{h \in \mathcal{H}}\sum_{t=1}^T \ell(h(x_t), y_t)$ to be the optimal hypothesis in hindsight.

For any bitstring $b \in \{0,1\}^T$, let $\phi : \{t \in [T] : b_t = 1\} \to \mathcal{S}_K$ denote a function mapping time points where $b_t = 1$ to permutations. Let $\Phi_b = \mathcal{S}_K^{\{t \in [T]:b_t=1\}}$ denote all such functions $\phi$. For every $h \in \mathcal{H}$, there exists a $\phi_b^h \in \Phi_b$ such that for all $t \in \{t : b_t = 1\}$, $\phi_b^h(t) = h(x_t)$. Let $|b| = |\{t \in [T] : b_t = 1\}|$. For every $b \in \{0,1\}^T$ and $\phi \in \Phi_b$, we will define an Expert $E_{b,\phi}$. Expert $E_{b,\phi}$, formally presented in Algorithm 3, uses $\mathcal{A}_i^p$'s to make predictions in each round. However, $E_{b,\phi}$ only updates the $\mathcal{A}_i^p$'s on those rounds where $b_t = 1$, using $\phi$ to compute a labeled instance. For every $b \in \{0,1\}^T$, let $\mathcal{E}_b = \bigcup_{\phi \in \Phi_b} \{E_{b,\phi}\}$ denote the set of all Experts parameterized by functions $\phi \in \Phi_b$. If $b$ is the bitstring with all zeros, then $\mathcal{E}_b$ will be empty. Therefore, we will actually define $\mathcal{E}_b = \{E_0\} \cup \bigcup_{\phi \in \Phi_b} \{E_{b,\phi}\}$, where $E_0$ is the expert that never updates $\mathcal{A}_i^j$'s and only uses them for predictions in all $t \in [T]$. Note that $1 \leq |\mathcal{E}_b| \leq (K!)^{|b|} \leq K^{K|b|}$. Using these experts, Algorithm 3 is our agnostic online learner $\mathcal{Q}$ for $\mathcal{H}$ w.r.t $\ell \in \mathcal{L}(\ell_{\text{prec}}^{@p})$.

---

**Algorithm 7** Expert $(b, \phi)$

**Input:** Independent copy of realizable learners $\mathcal{A}_i^p$ of $\mathcal{H}_i^p$ for $i \in [K]$

1 **for** $t = 1, ..., T$ **do**
2     Receive example $x_t$
3     Define a binary vote vector $v_t \in \{0,1\}^K$ such that $v_t[i] = \mathcal{A}_i^p(x_t)$
4     Predict $\hat{\pi}_t \in \arg\min_{\pi \in \mathcal{S}_K} \langle \pi, v_t \rangle$
5     **if** $b_t = 1$ **then**
6        Let $\pi = \phi(t)$ and for each $i \in [K]$, update $\mathcal{A}_i^p$ by passing $(x_t, \pi_i^p)$
7 **end**

---

Using REWA guarantees and following exactly the same calculation as in the proof of Theorem 5.1, we immediately arrive at

$$\mathbb{E}\left[\sum_{t=1}^T \ell(\mathcal{Q}(x_t), y_t)\right] \leq \mathbb{E}\left[\sum_{t=1}^T \ell(E_{B,\phi_B^{h^\star}}(x_t), y_t)\right] + M\sqrt{2T^{1+\beta}K\ln K},$$

the analog of Equation (1) for this setting. Using Lemma E.2, we have

$$\ell(E_{B,\phi_B^{h^\star}}(x_t), y_t) \leq \ell(h^\star(x_t), y_t) + \frac{M}{a}\ell(E_{B,\phi_B^{h^\star}}(x_t), \text{BinRel}(h^\star(x_t), p))$$

pointwise, where $a = \min_{\pi, y}\{\ell(\pi, y) \mid \ell(\pi, y) \neq 0\}$. By definition of $M$, we further get

$$\ell(E_{B,\phi_B^{h^\star}}(x_t), \text{BinRel}(h^\star(x_t), p)) \leq M\, \mathbb{1}\{\ell(E_{B,\phi_B^{h^\star}}(x_t), \text{BinRel}(h^\star(x_t), p)) > 0\}$$
$$= M\, \mathbb{1}\{\ell_{\text{prec}}^{@p}(E_{B,\phi_B^{h^\star}}(x_t), \text{BinRel}(h^\star(x_t), p)) > 0\},$$

where the equality follows from the fact that $\ell \in \mathcal{L}(\ell_{\text{prec}}^{@p})$.

In order to upperbound the indicator above, we need some more notations. Given the realizable online learner $\mathcal{A}_i^p$ for $i \in [K] \times [p]$, an instance $x \in \mathcal{X}$, and an ordered finite sequence of labeled examples $L \in (\mathcal{X} \times \{0,1\})^*$, let $\mathcal{A}_i^p(x|L)$ be the random variable denoting the prediction of $\mathcal{A}_i^p$ on the instance $x$ after running and updating on $L$. For any $b \in \{0,1\}^T$, $h \in \mathcal{H}$, and $t \in [T]$, let $L_{b<t}^h(i,p) = \{(x_s, h_i^p(x_s)) : s < t \text{ and } b_s = 1\}$ denote the *subsequence* of the sequence of labeled instances $\{(x_s, h_i^p(x_s))\}_{s=1}^{t-1}$ where $b_s = 1$. Then, we have

$$\mathbb{1}\{\ell_{\text{prec}}^{@p}(E_{B,\phi_B^{h^\star}}(x_t), \text{BinRel}(h^\star(x_t), p)) > 0\} \leq \sum_{i=1}^K \mathbb{1}\{\mathcal{A}_i^p(x_t \mid L_{B<t}^{h^\star}(i,p)) \neq h_i^{\star,p}(x_t)\}.$$

To prove this claimed inequality, consider the case when $\sum_{i=1}^K \mathbb{1}\{\mathcal{A}_i^p(x_t \mid L_{B<t}^{h^\star}(i,p)) \neq h_i^{\star,p}(x_t)\} = 0$ because the inequality is trivial otherwise. Then, we must have $\mathcal{A}_i^p(x_t \mid L_{B<t}^{h^\star}(i,p)) = h_i^{\star,p}(x_t)$ for all $i \in [K]$. Let $v_t \in \{0,1\}^K$ such that $v_t[i] = \mathcal{A}_i^p(x_t \mid L_{B<t}^{h^\star}(i,p))$ be a binary vote vector that the expert $E_{B,\phi_B^{h^\star}}$ constructs in round $t$. Since $h^\star(x_t)$ is a permutation, the vote vector $v_t$ must contain exactly $p$ labels with 1 vote and $K - p$ labels with 0 votes. Thus, every $\hat{\pi}_t \in \arg\min_{\pi \in \mathcal{S}_K} \langle \pi, v_t \rangle$ must rank labels with 1 vote in top $p$ and labels with 0 votes outside top $p$.

In other words, we must have $\hat{\pi}_t \overset{p}{=} h^\star(x_t)$, and thus $\ell_{\text{prec}}^{@p}(\hat{\pi}_t, \text{BinRel}(h^\star(x_t), p)) = 0$ by definition of $\ell_{\text{prec}}^{@p}$. Our claim follows because $E_{B,\phi_B^{h^\star}}(x_t) \in \arg\min_{\pi \in \mathcal{S}_K} \langle \pi, v_t \rangle$.

Combining everything, we obtain

$$\ell(E_{B,\phi_B^{h^\star}}(x_t), y_t) \leq \ell(h^\star(x_t), y_t) + \frac{M^2}{a} \sum_{i=1}^{K} \mathbb{1}\{\mathcal{A}_i^p(x_t \mid L_{B_{<t}}^{h^\star}(i,p)) \neq h_i^{\star,p}(x_t)\}.$$

Taking expectations on both sides and summing over all $t \in [T]$ yields

$$\mathbb{E}\left[\sum_{t=1}^{T} \ell(E_{B,\phi_B^{h^\star}}(x_t), y_t)\right] \leq \sum_{t=1}^{T} \ell(h^\star(x_t), y_t) + \frac{M^2}{a} \sum_{i=1}^{K} \mathbb{E}\left[\sum_{t=1}^{T} \mathbb{1}\{\mathcal{A}_i^p(x_t \mid L_{B_{<t}}^{h^\star}(i,p)) \neq h_i^{\star,p}(x_t)\}\right].$$

So, it now suffices to show that $\mathbb{E}\left[\sum_{t=1}^{T} \mathbb{1}\{\mathcal{A}_i^p(x_t \mid L_{B_{<t}}^{h^\star}(i,p)) \neq h_i^{\star,p}(x_t)\}\right]$ is a sub-linear function of $T$. Again, using the independence of $B_t$ and the algorithm's prediction in round $t$, we can write

$$\mathbb{E}\left[\sum_{t=1}^{T} \mathbb{1}\{\mathcal{A}_i^p(x_t \mid L_{B_{<t}}^{h^\star}(i,p)) \neq h_i^{\star,p}(x_t)\}\right] = \sum_{t=1}^{T} \mathbb{E}\left[\mathbb{1}\{\mathcal{A}_i^p(x_t \mid L_{B_{<t}}^{h^\star}(i,p)) \neq h_i^{\star,p}(x_t)\}\right] \frac{\mathbb{P}[B_t = 1]}{\mathbb{P}[B_t = 1]}$$

$$= \frac{T}{T^\beta} \sum_{t=1}^{T} \mathbb{E}\left[\mathbb{1}\{\mathcal{A}_i^p(x_t \mid L_{B_{<t}}^{h^\star}(i,p)) \neq h_i^{\star,p}(x_t)\}\right] \mathbb{E}\left[\mathbb{1}\{B_t = 1\}\right]$$

$$= \frac{T}{T^\beta} \sum_{t=1}^{T} \mathbb{E}\left[\mathbb{1}\{\mathcal{A}_i^p(x_t \mid L_{B_{<t}}^{h^\star}(i,p)) \neq h_i^{\star,p}(x_t)\} \mathbb{1}\{B_t = 1\}\right].$$

Next, we can use the regret guarantee of the algorithm $\mathcal{A}_i^p$ on the rounds it was updated. That is,

$$\sum_{t=1}^{T} \mathbb{E}\left[\mathbb{1}\{\mathcal{A}_i^p(x_t \mid L_{B_{<t}}^{h^\star}(i,p)) \neq h_i^{\star,p}(x_t)\} \mathbb{1}\{B_t = 1\}\right] = \mathbb{E}\left[\sum_{t:B_t=1} \mathbb{1}\{\mathcal{A}_i^p(x_t \mid L_{B_{<t}}^{h^\star}(i,p)) \neq h_i^{\star,p}(x_t)\}\right]$$

$$= \mathbb{E}\left[\mathbb{E}\left[\sum_{t:B_t=1} \mathbb{1}\{\mathcal{A}_i^p(x_t \mid L_{B_{<t}}^{h^\star}(i,p)) \neq h_i^{\star,p}(x_t)\} \,\middle|\, B\right]\right]$$

$$\leq \mathbb{E}_B\left[R_i^p(|B|)\right],$$

where $R_i^p(|B|)$ is the regret of $\mathcal{A}_i^p$, a sub-linear function of $|B|$. In the last step, we use the fact that $\mathcal{A}_i^p$ is a realizable algorithm for $\mathcal{H}_i^p$ and the feedback that the algorithm received was $(x_t, h_i^{\star,p}(x_t))$ in the rounds whenever $B_t = 1$. By Lemma 5.17 from Ceccherini-Silberstein et al. [2017], there exists a concave sub-linear function $\tilde{R}_i^p(|B|)$ that upperbounds $R_i^p(|B|)$. By Jensen's inequality, $\mathbb{E}_B\left[R_i^p(|B|)\right] \leq \tilde{R}_i^p(T^\beta)$, a sub-linear function of $T^\beta$.

Putting everything together, we obtain

$$\mathbb{E}\left[\sum_{t=1}^{T} \ell(\mathcal{Q}(x_t), y_t)\right] \leq \sum_{t=1}^{T} \ell(h^\star(x_t), y_t) + \frac{M^2}{a} \sum_{i=1}^{K} \frac{T}{T^\beta} \tilde{R}_i^p(T^\beta) + M\sqrt{2T^{1+\beta} K \ln K}$$

$$= \inf_{h \in \mathcal{H}} \sum_{t=1}^{T} \ell(h(x_t), y_t) + \frac{pM^2}{a} \sum_{i=1}^{K} \frac{T}{T^\beta} \tilde{R}_i^p(T^\beta) + M\sqrt{2T^{1+\beta} K \ln K}.$$

Since $\tilde{R}_i^p(T^\beta)$ is a sublinear function of $T^\beta$, we have that $\frac{T}{T^\beta} \tilde{R}_i^p(T^\beta)$ is a sublinear function of $T$. As the sum of sublinear functions is sublinear, the second term above must be a sublinear function of $T$. Thus, the regret is sub-linear for any choice of $\beta \in (0,1)$. This completes our proof as we have shown that the algorithm $\mathcal{Q}$ achieves sub-linear regret in $T$. $\qquad\square$

We will now show that the online learnability of $\mathcal{H}$ w.r.t $\ell$ implies that $\mathcal{H}_i^p$ for each $i \in [K]$ is online learnable w.r.t 0-1 loss.

*Proof.* (of necessity in Theorem 5.2)

Fix $\ell \in \mathcal{L}(\ell_{\text{prec}}^{@p})$ and let $M = \max_{\pi,y} \ell(\pi, y)$. Given an online learner $\mathcal{A}$ for $\mathcal{H}$ w.r.t $\ell$, our goal is to construct an agnostic online learner $\mathcal{A}_i^p$ for $\mathcal{H}_i^p$ for a fixed $i \in [K]$. One can construct agnostic online learners for $\mathcal{H}_i^p$ for all $i \in [K]$ by symmetry. Our construction uses the REWA and is similar to the sufficiency proof above.

Let us define function $\phi$'s, the collection of functions $\Phi_b$ for every $b$ in the same way we did before. For every $b \in \{0,1\}^T$ and $\phi \in \Phi_b$, define an Expert $E_{b,\phi}$. Expert $E_{b,\phi}$ is the expert presented in Algorithm 6 after setting $j = p$ and uses $\mathcal{A}$ to make predictions in each round. For every $b \in \{0,1\}^T$, let $\mathcal{E}_b = \bigcup_{\phi \in \Phi_b} \{E_{b,\phi}\}$ denote the set of all Experts parameterized by functions $\phi \in \Phi_b$. As before, we will actually define $\mathcal{E}_b = \{E_0\} \cup \bigcup_{\phi \in \Phi_b} \{E_{b,\phi}\}$, where $E_0$ is the expert that never updates $\mathcal{A}$ and only uses it to make predictions in each round. Note that $1 \leq |\mathcal{E}_b| \leq (K!)^{|b|} \leq K^{K|b|}$.

The online learner for $\mathcal{H}_i^p$, henceforth denoted by $\mathcal{Q}$, is similar to Algorithm 3. First, it samples a $B \in \{0,1\}^T$ s.t. $B_t \sim \text{Bernoulli}(T^\beta/T)$, constructs a set of experts $\mathcal{E}_B$ using Algorithm 6 and runs REWA, denoted by $\mathcal{P}$, on the 0-1 loss over the stream $(x_1, y_1), ..., (x_T, y_T) \in (\mathcal{X} \times \{0,1\})^T$. Let $h_i^{\star,p} = \arg\min_{h_i^p \in \mathcal{H}_i^p} \sum_{t=1}^T \mathbb{1}\{h_i^p(x_t) \neq y_t\}$ be the optimal function in hindsight and $h^\star$ be any arbitrary completion of $h_i^{\star,p}$.

Using REWA guarantees and following exactly the same calculation as in the sufficiency proof, we arrive at

$$\mathbb{E}\left[\sum_{t=1}^T \mathbb{1}\{\mathcal{Q}(x_t) \neq y_t\}\right] \leq \mathbb{E}\left[\sum_{t=1}^T \mathbb{1}\{E_{B,\phi_B^{h^\star}}(x_t) \neq y_t\}\right] + \sqrt{2T^{1+\beta} K \ln K}.$$

The inequality above is the adaptation of Equation (1) for this proof. Since $\mathbb{1}\{E_{B,\phi_B^{h^\star}}(x_t) \neq y_t\} \leq \mathbb{1}\{h_i^{\star,p}(x_t) \neq y_t\} + \mathbb{1}\{E_{B,\phi_B^{h^\star}}(x_t) \neq h_i^{\star,p}(x_t)\}$, the inequality above reduces to

$$\mathbb{E}\left[\sum_{t=1}^T \mathbb{1}\{\mathcal{Q}(x_t) \neq y_t\}\right] \leq \sum_{t=1}^T \mathbb{1}\{h_i^{\star,p}(x_t) \neq y_t\} + \mathbb{E}\left[\sum_{t=1}^T \mathbb{1}\{E_{B,\phi_B^{h^\star}}(x_t) \neq h_i^{\star,p}(x_t)\}\right] + \sqrt{2T^{1+\beta} K \ln K}.$$

It now suffices to show that $\mathbb{E}\left[\sum_{t=1}^T \mathbb{1}\{E_{B,\phi_B^{h^\star}}(x_t) \neq h_i^{\star,p}(x_t)\}\right]$ is sub-linear in $T$.

Given an online learner $\mathcal{A}$ for $\mathcal{H}$, an instance $x \in \mathcal{X}$, and an ordered finite sequence of labeled examples $L \in (\mathcal{X} \times \mathcal{Y})^*$, let $\mathcal{A}(x|L)$ be the random variable denoting the prediction of $\mathcal{A}$ on the instance $x$ after running and updating on $L$. For any $b \in \{0,1\}^T$, $h \in \mathcal{H}$, and $t \in [T]$, let $L_{b<t}^h = \{(x_i, \text{BinRel}(h(x_s), p)) : s < t \text{ and } b_s = 1\}$ denote the *subsequence* of the sequence of labeled instances $\{(x_s, \text{BinRel}(h(x_s), p))\}_{s=1}^{t-1}$ where $b_s = 1$. Using Lemma E.4, we have

$$\mathbb{1}\{E_{B,\phi_B^{h^\star}}(x_t) \neq h_i^{\star,p}(x_t)\} \leq \mathbb{1}\{\ell_{\text{prec}}^{@p}(\mathcal{A}(x_t \mid L_{B<t}^{h^\star}), \text{BinRel}(h^\star(x_t), p)) > 0\}$$
$$= \mathbb{1}\{\ell(\mathcal{A}(x_t \mid L_{B<t}^{h^\star}), \text{BinRel}(h^\star(x_t), p)) > 0\}$$
$$\leq \frac{1}{a} \ell(\mathcal{A}(x_t \mid L_{B<t}^{h^\star}), \text{BinRel}(h^\star(x_t), p)),$$

where the equality follows from the definition of the loss class. Here, $a$ is the lower bound on $\ell$ whenever it is non-zero. Thus, we obtain

$$\mathbb{E}\left[\sum_{t=1}^T \mathbb{1}\{E_{B,\phi_B^{h^\star}}(x_t) \neq h_i^{\star,p}(x_t)\}\right] \leq \frac{1}{a}\mathbb{E}\left[\sum_{t=1}^T \ell(\mathcal{A}(x_t \mid L_{B<t}^{h^\star}), \text{BinRel}(h^\star(x_t), p))\right]$$

Now, we will again use the fact that the prediction $\mathcal{A}(x_t \mid L_{B_{<t}}^{h^\star})$ only depends on $(B_1, \ldots, B_{t-1})$, but is independent of $B_t$. Using this independence, we obtain

$$\mathbb{E}\left[\sum_{t=1}^{T} \ell(\mathcal{A}(x_t \mid L_{B_{<t}}^{h^\star}), \mathrm{BinRel}(h^\star(x_t), p))\right] = \frac{T}{T^\beta} \mathbb{E}\left[\sum_{t:B_t=1} \ell(\mathcal{A}(x_t \mid L_{B_{<t}}^{h^\star}), \mathrm{BinRel}(h^\star(x_t), p))\right]$$

$$= \frac{T}{T^\beta} \mathbb{E}\left[\mathbb{E}\left[\sum_{t:B_t=1} \ell(\mathcal{A}(x_t \mid L_{B_{<t}}^{h^\star}), \mathrm{BinRel}(h^\star(x_t), p)) \Bigg| B\right]\right]$$

$$\leq \frac{T}{T^\beta} \mathbb{E}\left[R(|B|, K)\right],$$

where $R(|B|, K)$ is the regret of the algorithm $\mathcal{A}$ and is a sub-linear function of $|B|$. In the last step, we use the fact that $\mathcal{A}$ is a (realizable) online learner for $\mathcal{H}$ w.r.t. $\ell$ and the feedback that the algorithm received was $(x_t, \mathrm{BinRel}(h^\star(x_t), p))$ in the rounds whenever $B_t = 1$. Again, using Lemma 5.17 from Ceccherini-Silberstein et al. [2017] and Jensen's inequality yields $\mathbb{E}_B\left[R(|B|, K)\right] \leq \tilde{R}(T^\beta, K)$, a concave, sub-linear function of $T^\beta$. Combining everything, we get

$$\mathbb{E}\left[\sum_{t=1}^{T} \mathbb{1}\{\mathcal{Q}(x_t) \neq h_i^{\star,p}(x_t)\}\right] \leq \sum_{t=1}^{T} \mathbb{1}\{h_i^{\star,p}(x_t) \neq y_t\} + \frac{T}{a\,T^\beta} \tilde{R}(T^\beta, K) + \sqrt{2T^{1+\beta} K \ln K}$$

$$\leq \inf_{h_i^p \in \mathcal{H}_i^p} \sum_{t=1}^{T} \mathbb{1}\{h_i^p(x_t) \neq y_t\} + \frac{T}{a\,T^\beta} \tilde{R}(T^\beta, K) + \sqrt{2T^{1+\beta} K \ln K}$$

For any choice of $\beta \in (0, 1)$, the regret above is a sub-linear function of $T$. Therefore, we have shown that $\mathcal{Q}$ is an agnostic learner for $\mathcal{H}_i^p$ w.r.t. 0-1 loss. This completes our proof. $\qquad\square$

## E    Technical Lemmas

Throughout this section, for any ranking (permutation) $\pi \in \mathcal{S}_K$, we let $\pi_i^j = \mathbb{1}\{\pi_i \leq j\}$ for all $(i, j) \in [K]$.

**Lemma E.1.** *For any* $y \in \mathcal{Y}$, $(\pi, \hat{\pi}) \in \mathcal{S}_k$, *and* $\ell \in \mathcal{L}(\ell_{sum}^{@p})$

$$\ell(\pi, y) \leq \ell(\hat{\pi}, y) + c\,p\,\mathbb{E}_{j \sim Unif([p])}\left[\ell(\pi, BinRel(\hat{\pi}, j))\right].$$

*where* $c = \frac{\max_{\tilde{\pi}, y} \ell(\tilde{\pi}, y)}{\min_{\tilde{\pi}, y}\{\ell(\tilde{\pi}, y) \mid \ell(\tilde{\pi}, y) \neq 0\}}$.

*Proof.* Assume that $\ell(\pi, y) > \ell(\hat{\pi}, y) \geq 0$ (as otherwise the inequality trivially holds). Then, since $\ell \in \mathcal{L}(\ell_{sum}^{@p})$, it must be the case that $\hat{\pi} \overset{[p]}{\neq} \pi$. That is, $\hat{\pi}$ and $\pi$ assign different ranks to the labels in the top $p$. Therefore, there exists $i \in [p]$ s.t. $\ell_{sum}^{@p}(\pi, \mathrm{BinRel}(\hat{\pi}, i)) > 0$. Since $\ell \in \mathcal{L}(\ell_{sum}^{@p})$, for this same $i \in [p]$, $\ell(\pi, \mathrm{BinRel}(\hat{\pi}, i)) > 0$. Therefore, we have

$$c\,p\,\mathbb{E}_{j \sim \mathrm{Unif}([p])}\left[\ell(\pi, \mathrm{BinRel}(\hat{\pi}, j))\right] \geq c\ell(\pi, \mathrm{BinRel}(\hat{\pi}, i))$$

$$= \frac{\max_{\tilde{\pi}, y} \ell(\tilde{\pi}, y)}{\min_{\tilde{\pi}, y}\{\ell(\tilde{\pi}, y) \mid \ell(\tilde{\pi}, y) \neq 0\}} \ell(\pi, \mathrm{BinRel}(\hat{\pi}, i))$$

$$\geq \max_{\tilde{\pi}, y} \ell(\tilde{\pi}, y)$$

$$\geq \ell(\pi, y).$$

Combining the upperbounds in both cases gives the desired inequality. $\qquad\square$

**Lemma E.2.** *For any* $y \in \mathcal{Y}$, $(\pi, \hat{\pi}) \in \mathcal{S}_k$, *and* $\ell \in \mathcal{L}(\ell_{prec}^{@p})$

$$\ell(\pi, y) \leq \ell(\hat{\pi}, y) + c\,\ell(\pi, BinRel(\hat{\pi}, p)).$$

*where* $c = \frac{\max_{\tilde{\pi}, y} \ell(\tilde{\pi}, y)}{\min_{\tilde{\pi}, y}\{\ell(\tilde{\pi}, y) \mid \ell(\tilde{\pi}, y) \neq 0\}}$.

*Proof.* Assume that $\ell(\pi, y) > \ell(\hat{\pi}, y) \geq 0$ (as otherwise the inequality trivially holds). Then, since $\ell \in \mathcal{L}(\ell_{\text{prec}}^{@p})$, it must be the case that $\hat{\pi} \overset{p}{\neq} \pi$. That is, $\hat{\pi}$ and $\pi$ assign different labels in the top $p$. Therefore, $\ell_{\text{prec}}^{@p}(\pi, \text{BinRel}(\hat{\pi}, p)) > 0$. Since $\ell \in \mathcal{L}(\ell_{\text{prec}}^{@p})$, $\ell(\pi, \text{BinRel}(\hat{\pi}, p)) > 0$. Therefore, we have

$$
\begin{aligned}
c\,\ell(\pi, \text{BinRel}(\hat{\pi}, p)) &= \frac{\max_{\tilde{\pi}, y} \ell(\tilde{\pi}, y)}{\min_{\tilde{\pi}, y}\{\ell(\tilde{\pi}, y) \mid \ell(\tilde{\pi}, y) \neq 0\}} \ell(\pi, \text{BinRel}(\hat{\pi}, p)) \\
&\geq \max_{\tilde{\pi}, y} \ell(\tilde{\pi}, y) \\
&\geq \ell(\pi, y).
\end{aligned}
$$

Combining the upperbounds in both cases gives the desired inequality. $\qquad\square$

**Lemma E.3.** *Let $\pi, \hat{\pi} \in \mathcal{S}_k$. Then, for all $(i, j) \in [K] \times [p]$, $\ell_{sum}^{@p}(\pi, BinRel(\hat{\pi}, j)) \geq \mathbb{1}\{\pi_i^j \neq \hat{\pi}_i^j\}$.*

*Proof.* Fix label $i^\star \in [K]$ and threshold $j^\star \in [p]$. Our goal is to show that $\ell_{\text{sum}}^{@p}(\pi, \text{BinRel}(\hat{\pi}, j^\star)) \geq \mathbb{1}\{\pi_{i^\star}^{j^\star} \neq \hat{\pi}_{i^\star}^{j^\star}\}$. Recall that $\text{BinRel}(\hat{\pi}, j^\star)[i^\star] = \mathbb{1}\{\hat{\pi}_{i^\star} \leq j^\star\}$ by definition. Since $\ell_{\text{sum}}^{@p}(\hat{\pi}, \text{BinRel}(\hat{\pi}, j^\star)) = 0$, we have that

$$
\begin{aligned}
\ell_{\text{sum}}^{@p}(\pi, \text{BinRel}(\hat{\pi}, j^\star)) &= \ell_{\text{sum}}^{@p}(\pi, \text{BinRel}(\hat{\pi}, j^\star)) - \ell_{\text{sum}}^{@p}(\hat{\pi}, \text{BinRel}(\hat{\pi}, j^\star)) \\
&= \sum_{i=1}^{K} \min(\pi_i, p+1)\text{BinRel}(\hat{\pi}, j^\star)[i] - \sum_{i=1}^{K} \min(\hat{\pi}_i, p+1)\text{BinRel}(\hat{\pi}, j^\star)[i] \\
&= \sum_{i=1}^{K} \min(\pi_i, p+1)\mathbb{1}\{\hat{\pi}_i \leq j^\star\} - \sum_{i=1}^{K} \min(\hat{\pi}_i, p+1)\mathbb{1}\{\hat{\pi}_i \leq j^\star\} \\
&= \sum_{i=1}^{K} \min(\pi_i, p+1)\mathbb{1}\{\hat{\pi}_i \leq j^\star\} - \sum_{i=1}^{K} \hat{\pi}_i\mathbb{1}\{\hat{\pi}_i \leq j^\star\}
\end{aligned}
$$

Let $\mathcal{I} \subseteq [K]$ s.t. for all $i \in \mathcal{I}$, $\hat{\pi}_i^{j^\star} = \mathbb{1}\{\hat{\pi}_i \leq j^\star\} = 1$. Then, we have that

$$
\begin{aligned}
\ell_{\text{sum}}^{@p}(\pi, \text{BinRel}(\hat{\pi}, j^\star)) &= \sum_{i \in \mathcal{I}} \min(\pi_i, p+1) - \sum_{i \in \mathcal{I}} \hat{\pi}_i \\
&= \sum_{i \in \mathcal{I}} \min(\pi_i, p+1) - \sum_{i=1}^{j^\star} i
\end{aligned}
$$

Suppose that $\mathbb{1}\{\pi_{i^\star}^{j^\star} \neq \hat{\pi}_{i^\star}^{j^\star}\} = 1$. It suffices to show that $\ell_{\text{sum}}^{@p}(\pi, \text{BinRel}(\hat{\pi}, j^\star)) \geq 1$. There are two cases to consider. Suppose $i^\star \in \mathcal{I}$. Then, it must be the case that $\mathbb{1}\{\pi_{i^\star} \leq j^\star\} = \pi_{i^\star}^{j^\star} = 0$, implying that $\pi_{i^\star} \geq j^\star + 1$. It then follows that in the best case $\sum_{i \in \mathcal{I}} \min(\pi_i, p+1) \geq \sum_{i=1}^{j^\star-1} i + (j^\star + 1) > \sum_{i=1}^{j^\star} i$ showcasing that indeed $\ell_{\text{sum}}^{@p}(\pi, \text{BinRel}(\hat{\pi}, j)) \geq 1$. Now, suppose $i^\star \notin \mathcal{I}$. Then, $\mathbb{1}\{\hat{\pi}_{i^\star} \leq j^\star\} = 0$, which means that $\mathbb{1}\{\pi_{i^\star} \leq j^\star\} = 1$. Accordingly, while $\hat{\pi}$ did not rank label $i^\star$ in the top $j^\star$, $\pi$ *did* rank label $i^\star$ in the top $j^\star$. Since $|\mathcal{I}| = j^\star$, there must exist an label $\hat{i} \in \mathcal{I}$ which $\pi$ does not rank in the top $j^\star$. That is, there exists $\hat{i} \in \mathcal{I}$ s.t. $\pi_{\hat{i}} \geq j^\star + 1$. Using the same logic, in the best case $\sum_{i \in \mathcal{I}} \min(\pi_i, p+1) \geq \sum_{i=1}^{j-1} i + (j^\star + 1)$ showcasing that again $\ell_{\text{sum}}^{@p}(\pi, \text{BinRel}(\hat{\pi}, j^\star)) \geq 1$. Thus, we have shown that when $\mathbb{1}\{\pi_{i^\star}^{j^\star} \neq \hat{\pi}_{i^\star}^{j^\star}\} = 1$, $\ell_{\text{sum}}^{@p}(\pi, \text{BinRel}(\hat{\pi}, j^\star)) \geq 1$. Since $i^\star$ and $j^\star$ were arbitrary, this must be true for any $(i, j) \in [K] \times [p]$, completing the proof. $\qquad\square$

**Lemma E.4.** *Let $\pi, \hat{\pi} \in \mathcal{S}_k$. Then, for all $i \in [K]$, $\ell_{prec}^{@p}(\pi, BinRel(\hat{\pi}, p)) \geq \mathbb{1}\{\pi_i^p \neq \hat{\pi}_i^p\}$.*

*Proof.* Fix label $i^\star \in [K]$. Our goal is to show that $\ell^{@p}_{\text{prec}}(\pi, \text{BinRel}(\hat{\pi}, p)) \geq \mathbb{1}\{\pi^p_{i^\star} \neq \hat{\pi}^p_{i^\star}\}$. Recall that $\text{BinRel}(\hat{\pi}, p)[i^\star] = \mathbb{1}\{\hat{\pi}_{i^\star} \leq p\}$ by definition. Since $\ell^{@p}_{\text{prec}}(\hat{\pi}, \text{BinRel}(\hat{\pi}, p)) = 0$, we have that

$$\ell^{@p}_{\text{prec}}(\pi, \text{BinRel}(\hat{\pi}, p)) = \ell^{@p}_{\text{prec}}(\pi, \text{BinRel}(\hat{\pi}, p)) - \ell^{@p}_{\text{prec}}(\hat{\pi}, \text{BinRel}(\hat{\pi}, p))$$

$$= \sum_{i=1}^{K} \mathbb{1}\{\hat{\pi}_i \leq p\}\text{BinRel}(\hat{\pi}, p)[i] - \sum_{i=1}^{K} \mathbb{1}\{\pi_i \leq p\}\text{BinRel}(\hat{\pi}, p)[i]$$

$$= p - \sum_{i=1}^{K} \mathbb{1}\{\pi_i \leq p\}\mathbb{1}\{\hat{\pi}_i \leq p\}$$

Let $\mathcal{I} \subseteq [K]$ s.t. for all $i \in \mathcal{I}$, $\hat{\pi}^p_i = \mathbb{1}\{\hat{\pi}_i \leq p\} = 1$. Then, we have that

$$\ell^{@p}_{\text{prec}}(\pi, \text{BinRel}(\hat{\pi}, p)) = p - \sum_{i \in \mathcal{I}} \mathbb{1}\{\pi_i \leq p\}.$$

Suppose that $\mathbb{1}\{\pi^p_{i^\star} \neq \hat{\pi}^p_{i^\star}\} = 1$. It suffices to show that $\ell^{@p}_{\text{prec}}(\pi, \text{BinRel}(\hat{\pi}, p)) \geq 1$. There are two cases to consider. Suppose $i^\star \in \mathcal{I}$. Then, it must be the case that $\mathbb{1}\{\pi_{i^\star} \leq p\} = \pi^p_{i^\star} = 0$, implying that $\pi_{i^\star} \geq p + 1$. It then follows that in the best case $\sum_{i \in \mathcal{I}} \mathbb{1}\{\pi_i \leq p\} \leq p - 1 < p$ showcasing that indeed $\ell^{@p}_{\text{sum}}(\pi, \text{BinRel}(\hat{\pi}, p)) \geq 1$. Now, suppose $i^\star \notin \mathcal{I}$. Then, $\mathbb{1}\{\hat{\pi}_{i^\star} \leq p\} = 0$, which means that $\mathbb{1}\{\pi_{i^\star} \leq p\} = 1$. Accordingly, while $\hat{\pi}$ did not rank label $i^\star$ in the top $p$, $\pi$ *did* rank label $i^\star$ in the top $p$. Since $|\mathcal{I}| = p$, there must exist an label $\hat{i} \in \mathcal{I}$ which $\pi$ does not rank in the top $p$. That is, there exists $\hat{i} \in \mathcal{I}$ s.t. $\pi_{\hat{i}} \geq p + 1$. Using the same logic, in the best case $\sum_{i \in \mathcal{I}} \mathbb{1}\{\pi_i \leq p\} \leq p - 1 < p$ showcasing that again $\ell^{@p}_{\text{prec}}(\pi, \text{BinRel}(\hat{\pi}, p)) \geq 1$. Thus, we have shown that when $\mathbb{1}\{\pi^p_{i^\star} \neq \hat{\pi}^p_{i^\star}\} = 1$, $\ell^{@p}_{\text{prec}}(\pi, \text{BinRel}(\hat{\pi}, p)) \geq 1$. Since $i^\star$ was arbitrary, this must be true for any $i \in [K]$, completing the proof. $\qquad\square$