# OpenReview forum: "On the Learnability of Multilabel Ranking"
_NeurIPS.cc/2023/Conference — NeurIPS 2023 spotlight_

### Official Review · Reviewer_6zYz · 2023-07-04

**Soundness:** 3 good
**Presentation:** 2 fair
**Contribution:** 3 good
**Rating:** 6
**Confidence:** 2

**Summary:**

This paper is mainly about the learnability of multilabel ranking. The authors show that a ranking hypothesis class embeds $K^2$ different binary hypothesis classes and defines two families of ranking loss functions that capture most, if not all, ranking losses used in practice. The paper addresses the fundamental question of when a multilabel ranking problem is learnable, which remains unanswered despite the vast literature on multilabel ranking.

**Strengths:**

The paper provides a significant contribution to understanding when a multilabel ranking problem is learnable, addressing a fundamental question that has remained unanswered despite the extensive literature on multilabel ranking. The authors also provide sufficient conditions for the agnostic Probably Approximately Correct learnability of score-based ranking hypothesis classes.


**Weaknesses:**

This paper does not contain any experiments for their theories and algorithms.

**Questions:**

I have no questions about this paper.

**Limitations:**

This paper did not contain experiments for any proposed algorithm and losses.

---

> ### Author Rebuttal · Authors · 2023-08-07
>
> We thank the reviewer for noting that our work provides a significant contribution to the learnability of multilabel ranking problems.
>
> **Q**: *"This paper does not contain any experiments for their theories and algorithms"*
>
> A: We thank the reviewer for noting this concern. We would like to emphasize that the main focus of our work is theoretical as we aim to characterize the learnability of multilabel ranking problems. No prior characterization of learnability was known for multilabel ranking.  Developing ranking algorithms with strong practical guarantees has been extensively studied, but we believe it is out of scope for this paper.

---

### Official Review · Reviewer_ejP3 · 2023-07-06

**Soundness:** 3 good
**Presentation:** 3 good
**Contribution:** 3 good
**Rating:** 8
**Confidence:** 4

**Summary:**

This work study the learnability of multilabel ranking problems in both batch and online settings for different ranking losses family. They show that the multilabel ranking learnability is equivalent to the learnablities of a family of binary classification problems.

**Strengths:**

* The paper tackles a well-motivated problem. Besides, since the theory on the learnability of multilabel ranking is blank before this work, it seems that the novelity and originality of this paper can be promised.
*  The paper is overall well-written. The presentation of theoretical results is rigorous.
* The proofs are solid based my judgement.
* The results are surprisingly concise.
* The authors present a example stating that linear ranking hypothesis class is learnable, which shows that theri results are pratical.

**Weaknesses:**

As the authors confirmed, the bounds may be not optimal.

**Questions:**

N/A

---

> ### Author Rebuttal · Authors · 2023-08-07
>
> We thank the reviewer for finding that our paper tackles a well-motivated problem and presents a novel and original contribution to the theoretical foundations of multilabel ranking.

---

### Official Review · Reviewer_euAG · 2023-07-25

**Soundness:** 3 good
**Presentation:** 3 good
**Contribution:** 3 good
**Rating:** 7
**Confidence:** 3

**Summary:**

The authors in their work focus on a single theoretical contribution - showing that the multilabel ranking hypothesis class is PAC learnable for a wide range of ranking losses in both batch and online settings.
The authors first define two general families of ranking losses that are equivalent to popularly used losses.
 (like Pairwise Rank Loss, Discounted Cumulative Gain, Reciprocal Rank, Average Precision, Precision@p)
Then they show for the defined losses and both batch and online settings, the hypothesis class of multilabel ranking losses $\mathcal{H}$ can be expressed using binary hypothesis classes $\mathcal{H}_i^j$ for $i, j \in [K]$, where ($K$ is a number of labels), where $\mathcal{H}_i^j$ tels whether the label $i$ should be ranked in the top $j$ positions. The authors show that binary class $\mathcal{H}_i^j$ is PAC learnable, and as a result of that, it's also $\mathcal{H}$, and existing combinatorial dimensions for batch and online learning like VC and Littleston characterized learnability in the considered multilabel setting.

To be honest, I'm far from the best person to review this work, as I don't know related works that are the most important to this paper except the book of Shalev-Shwartz and Ben-David, which I read some time ago, so I cannot judge it in the context of recent works in this area. Since the proof is very slightly outlined in the main paper, I tried to check the appendix as much as time allowed me. I checked Appendix A, B, and most of C. I did not check Appendix D and E.

Assuming that all the parts are correct, I believe this is a solid contribution that should be accepted.

**Strengths:**

- The paper contributes to the fundamental understanding of statistical learning theory for learning multilabel ranking.
- The presented proofs are non-trivial but, at the same time, clearly presented in the appendix.

**Weaknesses:**

- The main proofs are only slightly outlined in the main paper; by slightly, I mean they don't give out any idea behind the proof, only the intermediate results. Because of that, the main paper feels more like an introduction to notation, and then it basically forces the reader to switch to reading a long appendix (or the read can just stop reading and believe the authors the result is valid). The paper would definitely benefit from a longer form. Also, the appendix would benefit from including the original theorems and lemmas to reduce the need for switching between the main paper and the appendix.
- It seems to me that two families of defined ranking losses capture "most if not all ranking losses used in practice" is a bit overstated. The result is not surprising as losses equivalent to these families (listed in Appendix A) are simple losses with (I think) the same form of an optimal hypothesis that predicts top $p$ or $K$ labels in descending order of expected relevance. These families may not capture more complex measures, but I agree that these are very popular.
- This is a notation-dense paper; as a reader, I would appreciate some more comments/full-sentence explanations on notation in Sections 2 and 3 to confirm my understanding of it before jumping to the main results. For example, it was not clear to me if relevance is descending, meaning B is the most relevant and 0 is the least relevant or its opposite. I could only confirm it by reading definitions of $\ell_{\text{sum}}^{@p}$, and $\ell_{\text{prec}}^{@p}$ and figuring out which one makes sense, but that was one page later. Again this I believe is a fault of limited space.
- NIT: The discussion section is not really a discussion, just a short summary.
- NIT: Lack of equation numbers is inconvenient when reading the paper like this (for example, to make notes on equations, as there is no defined way to address them).
- NIT: There are equations that go over the right margin, which is a bit unelegant when there is a strict space limit.
- NIT: Bit string? Why not just binary vector? To me bit string is a compact implementation of a binary vector.

**Questions:**

I don't really have questions.

**Limitations:**

There are no direct negative social impacts of this work. The limitation of tightness of sample complexity and regret is mentioned in the last section. I do not see any other limitations of this work.

---

> ### Author Rebuttal · Authors · 2023-08-07
>
> We thank the reviewer for finding that our paper contributes to the fundamental understanding of statistical learning theory for multilabel ranking.
>
> **Q1**: *"The main proofs are only slightly outlined in the main paper; "*
>
> A1: We thank the reviewer for pointing this out. We will make sure to add detailed proof sketches in the main text for the camera-ready version. In addition, we will include the original theorems and lemmas in the Appendix.
>
> **Q2**: *"It seems to me that two families of defined ranking losses capture "most if not all ranking losses used in practice" is a bit overstated. "*
>
> A2: We will dial back this claim and instead rephrase it to "many popular ranking losses used in practice".  That said, our loss families do capture most of the ranking losses used in practice. For example, all the ranking losses found in the popular MULAN library can be categorized into one of the 2 families.
>
> **Q3**: *"This is a notation-dense paper;"*
>
> A3: We agree with the reviewer that this is a notation-dense paper.  Will will make sure to add more explanations of the notation in Sections 2 and 3. Moreover, we will make sure to explicitly highlight that larger scores imply higher relevance.
>
> **Q4**: *"The discussion section is not really a discussion, just a short summary."*
>
> A4: We will expand on the discussion section by including limitations and future directions in the camera-ready version.
>
> **Q5**: *"NITS"*
>
> A5: We thank the reviewer for pointing out these issues. We will address all these comments in the camera-ready version.

---

> > ### Comment · Reviewer_euAG · 2023-08-14
> > **Re: Rebuttal by Authors**
> >
> > Thank you for your response.
> >
> > Regarding your response to Q2, I don't think MULAN lib is a good indicator/example here as it is no longer popular and widely used, and actively developed.
> >
> > Also, it would be nice to see a bit more complete answer to my Q4 and to Q4 of Reviewer edyo. Meaning to see at least a summary of these promised discussions.

---

> > > ### Author Response · Authors · 2023-08-14
> > > **Response to Reviewer euAG**
> > >
> > > Our discussion will roughly address the following points:
> > > 1. Our lower and upper bounds on the sample complexity and regret are not optimal. Given a particular ranking loss,  It will be interesting to derive the optimal sample complexity and regret in the realizable and agnostic settings. In addition, our bounds depend on the number of labels K. Recently, K-free bounds have been achieved for multiclass classification problems in both batch and online settings. An interesting future direction is to explore whether K-free bounds are possible for multilabel ranking.
> > > 2. Since our focus was on establishing learnability, our algorithms are not computationally efficient. However, in practice, people typically run ERM. So, tt is an interesting future direction to tightly quantify the sample complexity of ERM in the batch setting.
> > > 3. In learning theory, combinatorial dimensions play an important role in giving a tight quantitative characterization of learnability.  It is an interesting future direction to identify combinatorial dimensions that characterize multilabel ranking learnability for specific loss functions.
> > > 4. Our paper only studies a specific class of ranking loss functions. Accordingly, we leave it open to characterize the learnability of other natural ranking loss functions. Two such loss functions are recall@p and the thresholded version of the sum loss described in answer A7 of our respond to Reviewer edyo.
> > >
> > > With regards to consistency, we will expand on the following: there have been several works that have studied the consistency of convex surrogates of natural ranking losses such as the pairwise ranking loss, NDCG, Average Precision, and so forth [1, 2, 3, 4, 5, 6]. However, even for these aforementioned losses, the question of learnability has remained open. We close this gap by characterizing the learnability of these natural losses.
> > >
> > > Citations:
> > > 1. Duchi, John C., Lester W. Mackey, and Michael I. Jordan. "On the Consistency of Ranking Algorithms." ICML. 2010.
> > > 2. Buffoni, David, et al. "Learning scoring functions with order-preserving losses and standardized supervision." The 28th International Conference on Machine Learning (ICML 2011). 2011.
> > > 3. Gao, Wei, and Zhi-Hua Zhou. "On the consistency of multi-label learning." Proceedings of the 24th annual conference on learning theory. JMLR Workshop and Conference Proceedings, 2011.
> > > 4. Ravikumar, Pradeep, Ambuj Tewari, and Eunho Yang. "On NDCG consistency of listwise ranking methods." Proceedings of the Fourteenth International Conference on Artificial Intelligence and Statistics. JMLR Workshop and Conference Proceedings, 2011.
> > > 5. Calauzenes, Clément, Nicolas Usunier, and Patrick Gallinari. "On the (non-) existence of convex, calibrated surrogate losses for ranking." Advances in Neural Information Processing Systems 25 (2012).
> > > 6. Dembczynski, Krzysztof, Wojciech Kotlowski, and Eyke Hüllermeier. "Consistent multilabel ranking through univariate losses." arXiv preprint arXiv:1206.6401 (2012).

---

> > > > ### Comment · Reviewer_euAG · 2023-08-17
> > > > **Re: Response**
> > > >
> > > > Thank you for your further response,
> > > >
> > > > Regarding your summary of [1, 2, 3, 4, 5, 6]: I'm wondering if there are some more specific links between some of these works and yours, as consistency is kind of relaxed learnability.
> > > >
> > > > Nevertheless, after reading other reviews and your responses, I am going to retain my score.

---

### Official Review · Reviewer_edyo · 2023-07-27

**Soundness:** 3 good
**Presentation:** 2 fair
**Contribution:** 3 good
**Rating:** 6
**Confidence:** 3

**Summary:**

The authors study the problem of learnability of label ranking for a large family of ranking losses. The paper is purely theoretical. The main results have been built on the equivalence of realizable and agnostic learnability of the PAC model for binary classification. Using the techniques from Hopkins et al. (2022) and Raman et al. (2023), the authors show the learnability of label ranking in batch and online settings. Furthermore, they prove that ranking based on linear functions is learnable in the batch setting.


**Strengths:**

Strengths:
- Important theoretical contribution
- Practical learning problem

The paper is an important theoretical contribution for a practical learning problem. The result is novel and seems to be correct. Nevertheless, it is rather a straightforward extension of previous results.


**Weaknesses:**

Weak points:
- Confusing problem setup
- No discussion on related work
- Structure of the paper

The problem setting seems to be a bit confusing. I would call it rather label ranking (potentially with ties) instead of multi-label ranking. The latter suggests that the feedback information is binary. Alternatively, a wider discussion on differences between learning problems should be given.

The paper lacks discussion on related work. The obtained results should be discussed in light of the previous theoretical results on multi-label ranking (e.g., Dembczynski et al., 2012, Gao and Zhou, 2011, Koyejo et al., 2015). Unfortunately, the link between those results is not given in the paper.

Having a page limit, it is always hard to properly divide the results between the main text and the appendix. I appreciate that the authors decided to give the most important proofs in the main text. Unfortunately, the price for it is that the paper lacks the context (clear definition of the problem, discussion of ranking loss functions), wider discussion on the results (examples, related work), and discussion on limitations and practical implications.



**Questions:**

Learnability is an important concept, but consistency is important as well. How do the results from the submitted paper relate to the previous results concerning consistency of multi-label ranking?


**Limitations:**

The realizable-to-agnostic conversion is not computationally efficient. How does it impact the main findings of the submission?

It seems there is many other loss functions of "ranking" type in multi-label classification. How those results would apply to, for example, recall@p? It also seems that macro-metrics could also be interpreted as a kind of "ordering" functions. The abstract says that the results "capture most, if not all, losses used in practice." Is the statement not too strong? What are examples of losses not included in the two families considered in the paper?

---

> ### Author Rebuttal · Authors · 2023-08-07
>
> We thank the reviewer for noting that this work is an important theoretical contribution to a practical learning problem.
>
> **Q1**: *"The problem setting seems to be a bit confusing. I would call it rather label ranking (potentially with ties) instead of multi-label ranking. The latter suggests that the feedback information is binary."*
>
> A1: We thank the reviewer for pointing this out. We note that by taking B = 1 in Line 55, our model allows for the feedback to be binary. We will also include a broader discussion on the differences between learning problems in the camera-ready version.
>
> **Q2**: *"The paper lacks a discussion on related work."*
>
> A2: We thank the reviewer for pointing out these works. We will make sure to include a discussion of these works along with a comparison to our results in the camera-ready version.
>
> **Q3**: *"Unfortunately, the price for it is that the paper lacks the context (clear definition of the problem, discussion of ranking loss functions), wider discussion on the results (examples, related work), and discussion on limitations and practical implications."*
>
> A3: We thank the reviewer for bringing up these concerns. We will consider moving all proofs to Appendix and expand more on the problem setup, ranking loss functions, related work, limitations, and practical implications.
>
> **Q4**: *"Learnability is an important concept, but consistency is important as well."*
>
> A4: We thank the reviewer for pointing out the problem of consistency. While we do think that consistency is important, the main focus of this paper is establishing the learnability of multilabel ranking, which apriori, was not known. In the camera-ready version, we will make sure to reference and state the prior results on consistency for multi-label ranking and mention that the main focus of this work is on learnability.
>
> **Q5**: *"The realizable-to-agnostic conversion is not computationally efficient."*
>
> A5: We thank the reviewer for pointing this out. Indeed, our realizable-to-agnostic conversion is not computationally efficient. However, the main focus of this work is statistical and more specifically to provide a characterization of multilabel learnability. That said, we do think that establishing learnability via efficient reductions and constructing computationally efficient learning algorithms for multilabel ranking is an interesting future direction.
>
> **Q6**: *"It seems there is many other loss functions of "ranking" type in multi-label classification. How those results would apply to, for example, recall@p?"*
>
> A6: We thank the reviewer for pointing out that there are other loss functions of ranking type in multilabel classification, like the recall@p loss (under binary relevance score feedback). Unfortunately, the recall@p loss does not fall in either the sumloss@p family or the precision@p loss family. This is because if the number of relevant items is more than p, then recall@p will never be 0 (since it is not possible to return all the relevant items). In addition, the pairwise ranking loss is a popular "ranking" type loss in multi-label classification which we do capture in the sumloss family.  Overall, we agree that the loss functions captured by the sumloss@p and precloss@p families are not exhaustive. However, we do still believe that they capture the most popular ranking loss functions used in practice.
>
> **Q7**: *"The abstract says that the results "capture most, if not all, losses used in practice." Is the statement not too strong? What are examples of losses not included in the two families considered in the paper?"*
>
> A7: We will dial back this claim and instead rephrase it to "most ranking losses used in practice". That said, our loss families do capture most of the ranking losses used in practice. For example, all the ranking losses found in the popular MULAN library can be categorized into one of the 2 families. One example of a ranking loss that is not included in one of the two families is a thresholded version of the sum loss. That is, consider the loss function that is defined by first computing the sum loss and then setting it to zero if its value is smaller than some preselected threshold.

---

> > ### Comment · Reviewer_edyo · 2023-08-21
> > **After rebuttal**
> >
> > I thank the Authors for their responses. They in general answer my questions. Nevertheless, the paper needs substantial editorial work as the Authors promise to improve the text by adding additional clarifications, to move "all proofs to Appendix" and to "expand more on the problem setup," but also to "add detailed proof sketches in the main text" (as promised to Reviewer euAG). This is quite a challenge, but hope the Authors will succeed with it.

---

### Decision · Program_Chairs · 2023-09-21

**Decision:**

Accept (spotlight)

**Comment:**

The paper focuses on the learnability of multilabel ranking problems in both batch and online settings. The reviewers gave the paper a uniformly positive evaluation and generally find it to be a significant theoretical contribution to the field, addressing an important and previously unanswered question. There were some concerns and suggestions, mainly regarding clarity, related work discussion, and minor formatting issues, but they did not affect a positive reception of the paper.